# Evolution of cation binding in the active sites of P-loop nucleoside triphosphatases in relation to the basic catalytic mechanism

**Daria N Shalaeva**[1,2,3], **Dmitry A Cherepanov**[2,4], **Michael Y Galperin**[5], **Andrey V Golovin**[3], **Armen Y Mulkidjanian**[1,2,3]*

[1]School of Physics, University of Osnabrück, Osnabrück, Germany; [2]A.N. Belozersky Institute of Physico-Chemical Biology, Lomonosov Moscow State University, Moscow, Russia; [3]School of Bioengineering and Bioinformatics, Lomonosov Moscow State University, Moscow, Russia; [4]Semenov Institute of Chemical Physics, Russian Academy of Sciences, Moscow, Russia; [5]National Center for Biotechnology Information, National Library of Medicine, National Institutes of Health, Bethesda, United States

**\*For correspondence:**
amulkid@uos.de

**Abstract** The ubiquitous P-loop fold nucleoside triphosphatases (NTPases) are typically activated by an arginine or lysine 'finger'. Some of the apparently ancestral NTPases are, instead, activated by potassium ions. To clarify the activation mechanism, we combined comparative structure analysis with molecular dynamics (MD) simulations of Mg-ATP and Mg-GTP complexes in water and in the presence of potassium, sodium, or ammonium ions. In all analyzed structures of diverse P-loop NTPases, the conserved P-loop motif keeps the triphosphate chain of bound NTPs (or their analogs) in an extended, catalytically prone conformation, similar to that imposed on NTPs in water by potassium or ammonium ions. MD simulations of potassium-dependent GTPase MnmE showed that linking of alpha- and gamma phosphates by the activating potassium ion led to the rotation of the gamma-phosphate group yielding an almost eclipsed, catalytically productive conformation of the triphosphate chain, which could represent the basic mechanism of hydrolysis by P-loop NTPases.
DOI: https://doi.org/10.7554/eLife.37373.001

## Introduction

P-loop nucleoside triphosphatases (NTPases) represent the most common protein fold that can comprise up to 18% of all gene products in a cell. P-loop NTPase domains, which apparently preceded the Last Universal Cellular Ancestor, are found in translation factors, small GTPases, kinases, helicases, rotary ATP synthases, and many other ubiquitous proteins (*Koonin et al., 2000*; *Lupas et al., 2001*; *Leipe et al., 2002*; *Ponting and Russell, 2002*; *Anantharaman et al., 2003*; *Söding and Lupas, 2003*; *Orengo and Thornton, 2005*; *Ranea et al., 2006*; *Wittinghofer and Vetter, 2011*; *Verstraeten et al., 2011*; *Alva et al., 2015*; *Wuichet and Søgaard-Andersen, 2015*).

The P-loop fold, a variation of the Rossmann fold, is a 3-layer αβα sandwich, where the N-terminal β-strand is connected with the following α-helix by an elongated flexible loop typically containing the GxxxxGK[ST] sequence motif, known as the Walker A motif (*Walker et al., 1982*), see *Figure 1*. This motif is responsible for binding the NTP's phosphate chain and is often referred to also as the P-loop (phosphate-binding loop) motif (*Saraste et al., 1990*). The conserved lysine residue of the P-loop forms hydrogen bonds (H-bonds) with β- and γ-phosphate groups, while the next Ser/Thr

**Figure 1.** Mg-NTP complexes and their binding in the active sites of P-loop NTPases. Phosphate chains of NTP molecules and their analogs are colored by atoms: oxygen atoms in red, phosphorus in orange. The $K^+$ ion is shown as a purple sphere, $Na^+$ ion is shown as a blue sphere, $Mg^{2+}$ ions are shown as green spheres. Phosphate chain is shown in stick representation with oxygens in red and phosphorus atoms in orange; γ-phosphate mimicking groups ($AlF_4^-$ and $MgF_3^-$) are shown in black, coordination and hydrogen bonds are shown as black dashed lines. (**A**) Active site of the small Ras-like GTPase RhoA in complex with the activating protein RhoGAP [PDB entry 1OW3]; the bound GDP-$MgF_3^-$ mimics the transition state. The P-loop with the preceding α-helix is shown as green cartoon; Switch I motif with the conserved $Mg^{2+}$-binding Thr residue is shown in magenta; Switch II motif (DxxG motif, which starts from the conserved Asp of the Walker B motif) is shown in orange; the Arg finger of RhoGAP is colored turquoise. (**B**) Active site of the $K^+$-dependent GTPase MnmE with bound GDP-$AlF_4^-$ [PDB: 2GJ8]. Switch I region and the K-loop are shown in magenta. (**C**) The active site of dynamin, a $Na^+$-adapted GTPase with bound GDP-$AlF_4^-$ [PDB: 2X2E]. The P-loop and K-loop (Switch I region) are colored as in panels A and B. (**D**) Structure of the NTP triphosphate chain with $Mg^{2+}$ ion in a bidentate coordination, referred to as the βγ conformation. The pink dotted arch indicates the $P^B$-$O^{3B}$-$P^G$ angle; the blue dashed line indicates the $P^A$-$P^G$ distance. The atom names are in accordance with the CHARMM naming scheme (**Vanommeslaeghe et al., 2010**) and the recent IUPAC recommendations (**Blackburn et al., 2017**).

DOI: https://doi.org/10.7554/eLife.37373.002

residue coordinates the $Mg^{2+}$ ion, which, in its turn, coordinates β- and γ-phosphates from the other side of the phosphate chain (*Figure 1A–C*). Another motif typical for P-loop proteins is the Walker B motif with the sequence pattern *hhhh*D, where '*h*' denotes a hydrophobic residue (*Walker et al., 1982*). In P-loop NTPases, the aspartate from this motif either serves as a direct $Mg^{2+}$ ligand or participates in the second coordination sphere of $Mg^{2+}$ ion. Further specific motifs are shown in *Figure 1*.

Catalytic activity of P-loop NTPases typically depends upon their interaction with other proteins or domains of the same protein or RNA/DNA molecules; upon this interaction, activating Arg or Lys 'fingers' are inserted into the catalytic site (*Bos et al., 2007*), see *Figure 1A*. It still remains unclear whether there is some universal catalytic mechanism that is common for all P-loop NTPases, see (*Wittinghofer, 2006*; *Kamerlin et al., 2013*) for reviews. Recently, Gerwert and colleagues proposed that the Arg finger promotes GTP hydrolysis in small GTPases by rotating the α-phosphate with respect to β- and γ-phosphates towards an eclipsed conformation, which would favor the bond cleavage because of repulsion between the oxygen atoms of all three phosphate groups (*Rudack et al., 2012*; *Mann et al., 2016*; *Gerwert et al., 2017*). Blackburn and colleagues proposed that the insertion of the activating Arg residue leads to the reshuffling of the hydrogen bonded network, which drives the displacement of the attacking water molecule into the reactive position (*Jin et al., 2016*; *Jin et al., 2017a*; *Jin et al., 2017b*).

Some P-loop NTPases functionally depend not on Arg/Lys fingers, but on monovalent cations (*Figure 1B,C*, *Supplementary file 1A*). Strict dependence on $K^+$ ions was shown, among others, for the bacterial tRNA-modifying GTPase MnmE (also known as TrmE), Era-like GTPase Der from *Escherichia coli*, AtNOS/AtNOA1 GTPase from *Arabidopsis thaliana*, ribosome assembly GTPase YqeH, G-protein coupled with ferrous transporter FeoB, ribosome-binding ATPase YchF, bacterial ribosome biogenesis protein RgbA, and the DNA repair and recombination protein Rad51 (*Wu et al., 2005*; *Yamanaka et al., 2000*; *Scrima and Wittinghofer, 2006*; *Hwang and Inouye, 2001*; *Moreau et al., 2008*; *Anand et al., 2010*; *Ash et al., 2011*; *Ash et al., 2012*; *Tomar et al., 2011*; *Achila et al., 2012*; *Rice et al., 2001*; *Liu et al., 2004*; *Sehorn et al., 2004*; *Meyer et al., 2009*; *Böhme et al., 2010*). The requirement for $K^+$ or $NH_4^+$ ions was shown both for the intrinsic and ribosome-dependent GTPase activity of several ubiquitous translation factors (*Conway, 1964*; *Conway and Lipmann, 1964*; *Lubin and Ennis, 1964*; *Fasano et al., 1982*; *Ebel et al., 1992*; *Chinali and Parmeggiani, 1980*; *Dubnoff and Maitra, 1972*; *Kuhle and Ficner, 2014*). Based on the $K^+$-dependence of several ancient ATPases and GTPases of the TRAFAC class and on phylogenomic analysis, we have previously suggested that the dependence on $K^+$ ions was an ancestral trait which was subsequently replaced by the reliance on arginine or lysine fingers (*Mulkidjanian et al., 2012*; *Dibrova et al., 2015*).

In $K^+$-dependent P-loop NTPases, the catalytically important $K^+$ ion occupies the position of the positively charged nitrogen atom of the Arg/Lys finger, interacting with the phosphate groups of the NTP molecule from the opposite side of the $Mg^{2+}$ ion (*Scrima and Wittinghofer, 2006*; *Ash et al., 2012*), see also *Figure 1*. Using crystal structures of several $K^+$-dependent P-loop NTPases, a set of their characteristic features could be identified, including a specific $K^+$-binding 'K-loop' and two specific Asn/Asp residues in the P-loop (*Anand et al., 2010*; *Ash et al., 2012*; *Mulkidjanian et al., 2012*) (*Figure 1B,C*).

In the majority of $K^+$-dependent P-loop NTPases, $Na^+$ ions could not replace $K^+$ ions as cofactors (*Yamanaka et al., 2000*; *Hwang and Inouye, 2001*; *Moreau et al., 2008*; *Tomar et al., 2011*; *Achila et al., 2012*; *Böhme et al., 2010*). The very existence of ubiquitous $K^+$-dependent NTPases, along with the strict dependence of the translation system on cytoplasmic $K^+$ ions and its inhibition by $Na^+$ ions (*Lubin and Ennis, 1964*), require maintaining the $[K^+]/[Na^+]$ ratio >>1.0 in the cytoplasm. Since $Na^+$ usually prevails over $K^+$ in natural habitats, cells may spend up to a half of the available energy to maintain the proper $[K^+]/[Na^+]$ ratio (*Skulachev, 1978*). It has been argued that the first cells emerged in $K^+$-rich environments, which could explain the $K^+$ dependence of the evolutionarily old cellular processes (*Mulkidjanian et al., 2012*; *Dibrova et al., 2015*). However, it has remained obscure why, in the course of evolution, the cellular machinery has not switched its specificity from $K^+$ to $Na^+$, considering the abundance of $Na^+$ in natural habitats (*Drever, 1997*). Such adaptation would have been widely beneficial, especially in the case of marine organisms, which invest large efforts into counteracting the $[K^+]/[Na^+]$ ratio of ~0.02 in the sea water (*Oren, 2011*). For P-loop NTPases, the use of $Na^+$ ion as an activating cofactor is, in principle, possible: human

dynamin and the dynamin-like protein from *A. thaliana* are equally well activated by $Na^+$ and $K^+$ ions (*Chappie et al., 2010*; *Yan et al., 2011* ). The structures of dynamins show that $Na^+$ ions bind in a similar position to that occupied by $K^+$ ions in potassium-dependent NTPases (*Ash et al., 2012*), *cf.* *Figure 1B and C*. Therefore, the strong preference of other NTPases for $K^+$ ions remains a mystery.

The specific role of $K^+$ ions in processing phosphoanhydride bonds has been documented also in the absence of enzymes. Back in 1960, larger ions, such as $K^+$ and $Rb^+$, were shown to be more efficient than the smaller $Na^+$ and $Li^+$ ions in accelerating transphosphorylation (*Lowenstein, 1960*), see *Supplementary file 1B*. These observations suggested that the observed catalytic effect of the positive charges of Arg/Lys fingers or $K^+$ ions could be determined by the size of these cations.

Here, we have performed evolutionary analysis of the conformations of NTPs and their analogs bound in the active sites of different families of P-loop NTPases and complemented this analysis with molecular dynamics (MD) simulations. We report that, in MD simulations of the Mg-NTP complexes in water in the presence of $K^+$, $Na^+$, and $NH_4^+$ ions, these ions (hereafter $M^+$ ions) got bound to the phosphate chain in the same two sites that are taken by positive charges in the active sites of P-loop NTPases, namely, (i) between β- and γ-phosphates, in the position of the amino group of the invariant P-loop lysine residue, and (ii) between α- and γ-phosphates, in the position that is occupied either by the side chain of the activating Arg/Lys finger or by an $M^+$ ion. The extended conformation of the phosphate chain, which is similar to the catalytically prone conformation of tightly bound Mg-NTP complexes in the active sites of diverse P-loop NTPases, was achieved in water only in the presence of the larger $K^+$ and $NH_4^+$ ions, but not with the smaller $Na^+$ ions. The MD simulations of the $K^+$-dependent GTPase MnmE showed that binding of the activating $K^+$ ion between α- and γ-phosphates of an extended triphosphate chain led to the rotation of the γ-phosphate group yielding an almost eclipsed, catalytically productive conformation of the triphosphate chain. In addition, comparative structural analysis has revealed that, although the activating $M^+$ ions are bound exclusively by the residues of the P-loop NTPase domain, the activation of respective NTPases additionally requires a specific interaction of the P-loop domain with the respective activating moiety (another protein domain or an RNA/DNA molecule) to shape the cation-binding site. Such a mechanism prevents uncontrolled hydrolysis of the cellular NTP stock, which, otherwise, could cause cell death.

## Results

### Molecular dynamics simulations of $Mg^{2+}$-NTP complexes in water and comparison with structures of P-loop NTPases

#### Molecular dynamics simulations

We have conducted a series of molecular dynamics (MD) simulations of the $Mg^{2+}$-ATP and $Mg^{2+}$-GTP complexes (hereafter Mg-ATP and Mg-GTP, respectively) in water and in the presence of $K^+$, $Na^+$, or $NH_4^+$ ions (see Materials and methods and *Supplementary file 1C* for details). To our knowledge, no computational studies of Mg-NTP complexes investigated the effects of monovalent cations.

As a starting point for the MD simulations, we chose the conformation of Mg-ATP complex with the $Mg^{2+}$ ion coordinated by $O^{1B}$ and $O^{2G}$ oxygen atoms of the β- and γ-phosphate groups and four water molecules (hereafter, the atom names follow the CHARMM naming scheme (*Vanommeslaeghe et al., 2010*) and the recent IUPAC Recommendations (*Blackburn et al., 2017*), as shown in *Figure 1D*). This mode of $Mg^{2+}$ coordination, often referred to as bidentate or βγ coordination, has been observed in NMR studies of the Mg-ATP complex in water (*Cohn and Hughes, 1962*; *Jiang and Mao, 2002*; *Huang and Tsai, 1982*; *Cowan, 1991*) and in crystal structures of P-loop NTPases with bound NTPs and their analogs (*Saraste et al., 1990*; *Abrahams et al., 1994*; *Schweins and Wittinghofer, 1994*; *Scheffzek et al., 1997*), see also *Figure 1*. The initial structure of the Mg-ATP complex was optimized in vacuum using the PM3 Hamiltonian. After that, 1200 water molecules and six monovalent cations ($K^+$, $Na^+$ or $NH_4^+$) were added to the Mg-ATP complex. In each case, 4 $Cl^-$ ions were added to balance the total charge of the simulation system. The resulting solution corresponded to the total ionic strength of 0.2 M. To investigate the conformational space of the Mg-ATP complex in water, we performed three independent MD simulation runs of 170 ns for each system. During each simulation, the system coordinates were saved every 50 picoseconds.

Since we were mostly interested in the βγ conformations of the Mg-ATP complex that are typical for P-loop NTPases, we have conducted an additional series of 25 independent 20-ns long MD simulations, with and without M$^+$ ions to sample enough βγ conformations for comparative analyses. The simulations were performed both for Mg-ATP and Mg-GTP complexes (*Supplementary file 1C*). These data were used for analyses of conformations of triphosphate chain in the presence of different M$^+$ ions. Generally, the results were very similar for the Mg-ATP and Mg-GTP complexes, therefore hereafter we describe only the Mg-ATP data. The MD simulation data for Mg-GTP complexes are presented as figure supplements and referred to where appropriate.

## Cation binding to Mg$^{2+}$-NTP complexes in water

Distance distributions obtained from the MD simulation data (*Figure 2*, *Figure 2—figure supplement 1*) show that M$^+$ ions formed coordination bonds with oxygen atoms of the ATP phosphate chain with the respective lengths of 2.2 Å for Na$^+$, 2.6 Å for K$^+$, and 2.7 Å for NH$_4^+$ ions. These distances correspond well with the crystallographic data for these ions (*Harding, 2002*; *Harding, 2004*; *Sigel et al., 2016*). On time average, within the 4 Å radius around the phosphate chain, 1.5 cations were present in the case of Na$^+$ and NH$_4^+$, and 0.75 cations were present in the case of K$^+$ (*Figure 2—figure supplement 2*). Based on the radial distributions of M$^+$ ions around each individual oxygen atom of the ATP phosphate chain (*Figure 2—figure supplement 1*) and visual inspection of the M$^+$ binding to the phosphate groups, at least two distinct binding sites for M$^+$ ions could be identified (*Figure 2A*). One of them was formed by the oxygen atoms of β- and γ-phosphates, and the other site involved the oxygens of α- and γ-phosphates. We refer to these binding sites as the BG and AG sites, respectively. Additionally, M$^+$ ions were often found close to the distal end of the phosphate chain, where they contacted one or more oxygen atoms of the γ-phosphate (the G site(s), *Figure 2A*).

To characterize M$^+$ binding in the AG and BG sites, we measured the distances from each M$^+$ ion to the nearest oxygen atoms of the two respective phosphate residues (R$^{AG}$ and R$^{BG}$ distances in *Figure 2B*). Site occupancy was estimated, as shown in *Figure 2C–E*, from the number of M$^+$ ions located in the proximity of the binding site at each moment of the simulation. In the BG site, binding of any M$^+$ ion produced a prominent maximum in the R$^{BG}$ distribution. The R$^{BG}$ values peaked at the same distance as the maxima of the distribution of distances to separate oxygens (*Figure 2—figure supplement 1*), which indicates that the cations in the BG site simultaneously formed coordination bonds with two oxygen atoms. Similarly, in the AG site, the NH$_4^+$ and Na$^+$ ions produced peaks in the R$^{AG}$ distribution plots with the maxima at 2.7 Å and 2.3 Å, respectively. For K$^+$ ions, the corresponding peak with a R$^{AG}$ value of 2.6 Å was wide. Still, the distributions of the distances between cations and individual oxygen atoms of the triphosphate chain show that oxygen atoms of γ-phosphate had the most contacts with K$^+$ ions, see graphs in *Figure 2—figure supplement 1*.

While occupying the same binding sites, M$^+$ ions bound with different affinity that decreased in the order of Na$^+$ > NH$_4^+$ > K$^+$ (*Supplementary file 1B*). Higher affinity of ATP to Na$^+$ ions, as compared to K$^+$ and NH$_4^+$ ions, was previously observed in several experimental studies, albeit in the absence of Mg$^{2+}$ (*Supplementary file 1B*). For each M$^+$ ion, MD simulation data indicated much lower occupancy of the AG site than of the BG site; the average occupancy of the BG site was estimated to be 0.95 for Na$^+$, 0.72 for NH$_4^+$, and 0.5 for K$^+$, compared to the average occupancy of the AG site of 0.15 for Na$^+$, 0.2 for NH$_4^+$, and 0.05 for K$^+$ (*Figure 2C–E*).

In MD simulations of Mg-GTP complexes, the M$^+$ binding pattern was similar to that of Mg-ATP complexes, see *Figure 2—figure supplement 3*.

The reasons for the weak K$^+$-binding in the AG site could be, in principle, clarified by structural and thermodynamic analysis of the conformations of the Mg-NTP complex with two bound K$^+$ ions. Such an analysis, however, was hindered by the scarcity of the respective MD simulation frames. Therefore, we have conducted additional MD simulations with positional restraints applied to the cations (*Supplementary file 1C*). We have conducted 10-ns simulations of an ATP molecule with Mg$^{2+}$ in the βγ coordination and K$^+$ in the BG site, and of the same system but with the addition of the second K$^+$ ion in the AG site. Positional restrains were applied to K$^+$ and Mg$^{2+}$ ions and to one of the atoms of the adenine base. Binding of the second K$^+$ ion in the AG site was found to stabilize all three phosphate groups in a near-eclipsed conformation, with the phosphorus-oxygen bonds of the α-phosphate group almost coplanar to the respective bonds of β- and γ-phosphates

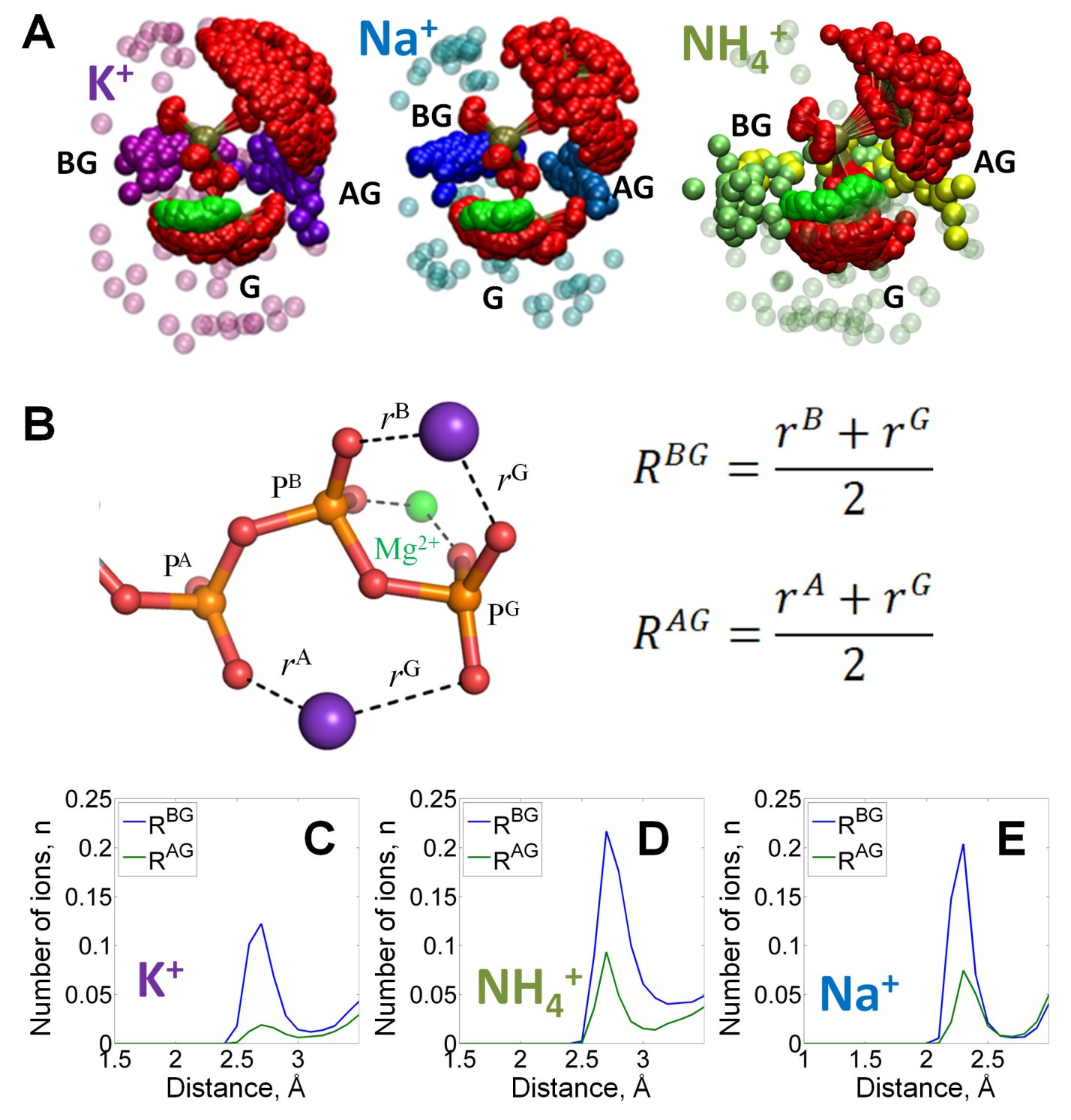

**Figure 2.** Binding of monovalent cations to the Mg-ATP in water. The color scheme is as in *Figure 1*. (**A**) Superposition of the ATP phosphate chain conformations observed in the MD simulations in the presence of K+ ions (shown in purple); Na+ ions (shown in blue) and NH4+ ions (nitrogen atoms of NH4+ ions are shown in yellow/green). The ribose and adenine moieties are not shown, the phosphate chain is shown with PA on top and PG at the bottom. All cations within 5 Å from the phosphate chain are shown and colored in different shades depending on the nearby oxygen atoms to illustrate the distinction between binding in the AG and BG sites (see text for details). Transparent spheres signify the ions outside the AG and BG sites. The constellation of ions in the vicinity of γ-phosphate is referred to as the site G. For the visualization, we have selected every 100th simulation frame to sample the conformational states of the Mg-ATP complex with 5-ns intervals. The conformations were superposed to achieve the best possible match

*Figure 2 continued on next page*

Figure 2 continued
between coordinates of the phosphorus and ester oxygen atoms of the ATP phosphate chain. (B) Geometry of the Mg-ATP complex with two monovalent cations bound, one in the AG site and one in the BG site. Distances to the AG and BG binding sites ($R^{AG}$ and $R^{BG}$) were calculated as averages of the distances to the two corresponding oxygen atoms. The distances to the oxygen atoms (e.g. $r^A$) were defined as the shortest distances between a particular $M^+$ ion and any oxygen atom of the respective phosphate group (including the bridging oxygen atoms). (C-E) distance distributions for $K^+$, $NH_4^+$, and $Na^+$ ions in the AG and BG sites.

DOI: https://doi.org/10.7554/eLife.37373.003

The following figure supplements are available for figure 2:

**Figure supplement 1.** Radial distribution of cations in the proximity of each oxygen atom.
DOI: https://doi.org/10.7554/eLife.37373.004
**Figure supplement 2.** Properties of cation binding to the ATP as derived from MD simulations.
DOI: https://doi.org/10.7554/eLife.37373.005
**Figure supplement 3.** Binding of monovalent cations to the Mg-GTP in water.
DOI: https://doi.org/10.7554/eLife.37373.006

(*Figure 3*; *Supplementary file 1D*). Only in this conformation, the distance between the oxygen atoms of α- and γ-phosphates was short enough to accommodate the second $K^+$ ion. As shown in *Figure 3—figure supplement 1*, binding of the second $K^+$ ion in the AG site promotes the transition of the phosphate chain into the almost fully eclipsed conformation by approximately 27 meV or 2.5 kJ/mol.

## Shape of the triphosphate chain of $Mg^{2+}$-NTP complexes in water as inferred from the MD simulation data

Cleavage of the bond between β- and γ-phosphates is believed to proceed via a planar transition complex, whereby the $P^B$-$O^{3B}$-$P^G$ angle widens (*Kamerlin et al., 2013*; *Jin et al., 2017a*; *Jin et al., 2017b*; *Kiani and Fischer, 2016*; *Warshel and Prasad, 2017*; *Akola and Jones, 2003*; *Grigorenko et al., 2006*; *Harrison and Schulten, 2012*). Another important feature of the Mg-ATP complex is the curvature of the phosphate chain, which can be characterized by the $P^A$-$P^G$ distance (*Figure 1D*). During all MD simulations, $P^A$-$P^G$ distances and $P^B$-$O^{3B}$-$P^G$ angles fluctuated around a certain value for a while and then switched to another set of values; this behavior reflected periods of MD trajectories characterized by the same type of interaction between the $Mg^{2+}$ ion and the tri-phosphate chain (*Figure 4* and *Figure 4—figure supplement 1*). The ATP molecules switched between the bidentate βγ conformation and the so-called αβγ conformations with the $Mg^{2+}$ ion being coordinated by one oxygen atom from each phosphate group (tridentate coordination of $Mg^{2+}$). The latter conformation is known from $^{31}P$ NMR studies (*Huang and Tsai, 1982*; *Mildvan, 1987*) and some proteins (*Chaudhry et al., 2003*; *Wang and Boisvert, 2003*). In the long (3 × 170 ns) simulations, several versions of the αβγ conformation could be seen, differing in the particular oxygen atoms of the phosphate chain that were involved in the tridentate coordination of the $Mg^{2+}$ ion (*Figure 4—figure supplement 1*). In short, MD simulations that started from the same βγ conformation (simulations 5–8 in *Supplementary file 1C*), we did not observe significant differences in the lifetimes of the βγ conformation between systems with different cations (*Supplementary file 1E*).

We used the values of $P^A$-$P^G$ distances and $P^B$-$O^{3B}$-$P^G$ angles as parameters to describe the geometry of the ATP phosphate chain in the presence and absence of different $M^+$ ions. In each of the sampled conformations, the Mg-ATP complex was characterized by distinct $P^A$-$P^G$ distances and $P^B$-$O^{3B}$-$P^G$ angles, which depended on the nature of the added monovalent cation (*Figure 4*, *Table 1*). While all $M^+$ ions seemed to contract the phosphate chain, it was more extended in the presence of $K^+$ ions than in the presence of $NH_4^+$ or $Na^+$ ions. Furthermore, $Na^+$ and $NH_4^+$ ions could induce an even more compressed, curled conformation of the Mg-ATP complex with even shorter distances between $P^A$ and $P^G$ atoms. Such curled conformations of the phosphate chain were not observed either in the presence of $K^+$ ions or in the absence of $M^+$ ions (*Figure 4*, *Table 1*).

*Figure 5* shows heat maps of the conformations seen in the MD simulations with the values of $P^B$-$O^{3B}$-$P^G$ angle and $P^A$-$P^G$ distance used as coordinates. The shading reflects the probability (normalized frequency) of conformations corresponding to the respective measurements. For the βγ

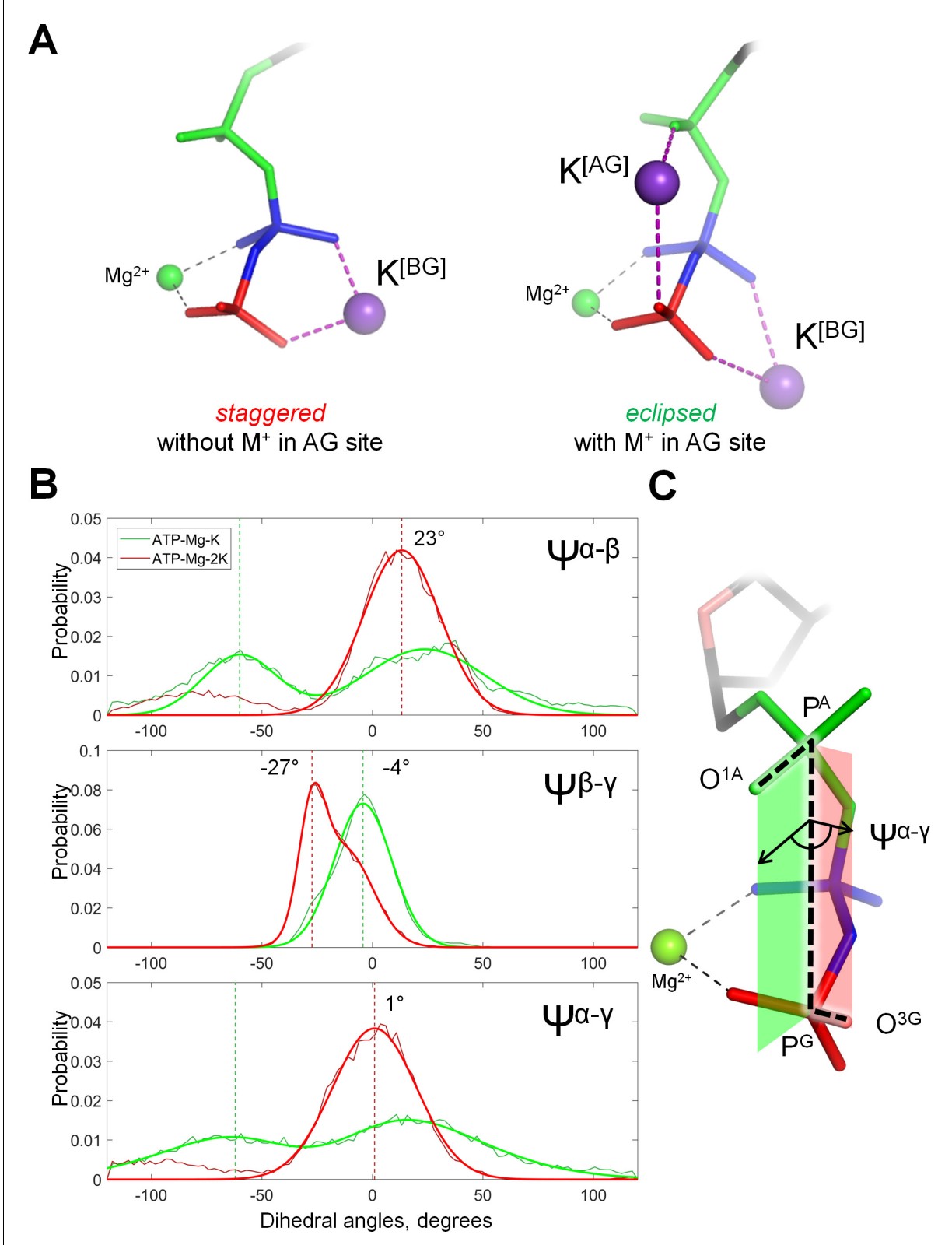

**Figure 3.** Cation binding induces eclipsed conformation of the phosphate chain. (**A**) Conformations of Mg-ATP complexes with one and two K$^+$ ions bound as inferred from MD simulations; left structure, no K$^+$ ion bound in the AG site; right structure, a K$^+$ ion is bound in the AG site. The α-phosphate is in on the top, β- and γ-phosphates are below; the α-phosphate is shown in green, β-phosphate in blue, γ-phosphate in red. (**B**) Distribution histograms for dihedral angles between phosphate groups in ATP, calculated from MD simulations of Mg-ATP with one K$^+$ cation bound in

*Figure 3 continued on next page*

*Figure 3 continued*

the BG site (green) and with two cations bound in the AG and BG sites (red). Normalized histograms of dihedral angle distribution (thin lines) were calculated from MD trajectories and fitted with normal distribution function (thick lines). Dashed lines indicate the centroid values of the fits by Gaussian function. All distributions were fitted with one-term Gaussian models, except for the $\Psi^{\beta-\gamma}$ angle in case of Mg-ATP with two cations bound, this distribution was fitted with a two-term Gaussian, parameters for the highest peak are shown. (C) The phosphate chain of GTP, illustrating the dihedral angle $\Psi^{\alpha-\gamma}$. Dihedral angle is an angle between two planes and is defined by four atoms. In this case, the angle $\Psi^{\alpha-\gamma}$ is an angle between the plane that contains atoms $P^G$, $P^A$ and $O^{1A}$ (green), and the plane that contains atoms $P^A$, $P^G$ and $O^{3G}$ (red). In the fully eclipsed conformation, both P-O bonds are coplanar, so that the two planes overlap and the dihedral angle between them is 0°.

DOI: https://doi.org/10.7554/eLife.37373.007

The following figure supplement is available for figure 3:

**Figure supplement 1.** Coupling between cation binding in the AG site and rotation of γ-phosphate relative to α- and β-phosphates.

DOI: https://doi.org/10.7554/eLife.37373.008

conformation of the Mg-ATP complex, the largest $P^A$-$P^G$ distances, up to 5.5 Å, were observed in simulations without $M^+$ ions (*Figures 4* and *5*). Presence of $M^+$ ions in the simulation system led to a significant decrease of the $P^A$-$P^G$ distances (*Figures 4* and *5*, *Table 1*). The $P^B$-$O^{3B}$-$P^G$ angles in the βγ-coordinated Mg-ATP complexes did not differ significantly between simulations with different cations or without cations added (*Figures 4* and *5*, *Table 1*). Among the studied cations, $K^+$ ions allowed for the longest $P^A$-$P^G$ distances.

The shapes of triphosphate chains of Mg-GTP complexes in water in the presence of the studied $M^+$ ions were very similar to those of Mg-ATP complexes, see *Figure 5—figure supplement 1*.

## Shape of the phosphate chain in the structures of P-loop NTPases

Binding in the catalytic site of a P-loop NTPase imposes constraints on the Mg-NTP complex, so that only particular conformations of the phosphate chain are allowed. These conformations appear to be catalytically prone, since NTP binding to an inactive P-loop domain (in the absence of a specific activating protein) already increases the rate constant of hydrolysis by several orders of magnitude as compared to the NTP hydrolysis in water (*Kötting and Gerwert, 2004*; *Shutes and Der, 2006*).

We analyzed the shapes of phosphate chains and the positions of positive charges around them in the available crystal structures of P-loop NTPases and compared them with the topology of Mg-ATP complexes seen in our MD simulations. The InterPro (*Finn et al., 2017*) entry for 'P-loop containing nucleoside triphosphate hydrolase' (IPR027417) listed 2,899 X-ray and 55 solution NMR structures of P-loop proteins. From this list, we selected those X-ray structures that contain $Mg^{2+}$ ion and an NTP-like molecule located in the proximity of at least one Lys residue, which would indicate that this NTP-like molecule is bound in the active site. Using these criteria, we identified 671 Protein Data Bank (PDB) entries, many of them with multiple subunits, resulting in the total of 1,357 Mg-NTP-like complexes. Crystal structures with non-hydrolyzable NTP analogs were used to gather information on the shape(s) of the phosphate chain in a potentially catalytically-prone conformation (s). In structures with transition state analogs, the $AlF_3$/$BeF_3^-$/$MgF_3^-$ or $AlF_4^-$ moieties mimicked the γ-phosphate group (*Jin et al., 2017b*; *Wittinghofer, 1997*; *Menz et al., 2001*). These structures were used as closest approximations of the nucleotide conformations in the transition state.

To characterize the conformations of the phosphate chain in the active sites of P-loop proteins, we used the same parameters as for the MD simulation data, namely the $P^A$-$P^G$ distance (or the corresponding distances in substrate analogs) and the value of the $P^B$-$O^{3B}$-$P^G$ angle (or the corresponding angles in substrate analogs). Using these two parameters as coordinates, we mapped the conformations attained by NTP-like molecules in the crystal structures (separately shown and described in *Figure 5—figure supplement 2*) on the heat maps for all four systems, calculated from MD simulations (*Figure 5*, *Figure 5—figure supplement 1*). In the top row of *Figure 5*, the heat maps included all conformations of Mg-ATP in water, including those not found in crystal structures of P-loop NTPases, for example with αβγ coordination of $Mg^{2+}$, as shown in *Figure 4*. Therefore, conformations of Mg-ATP complexes from MD simulations only partially overlapped with the conformations of non-hydrolyzable analogs of NTPs in P-loop NTPases (the blue contours in *Figure 5A*). The extent of the overlap depended on the nature of the cation used in MD simulations: it was highest with $K^+$ and lowest with $Na^+$. The extent of this overlap was less when the data from MD simulations were compared to the conformations of transition state analogs (the pink contours in

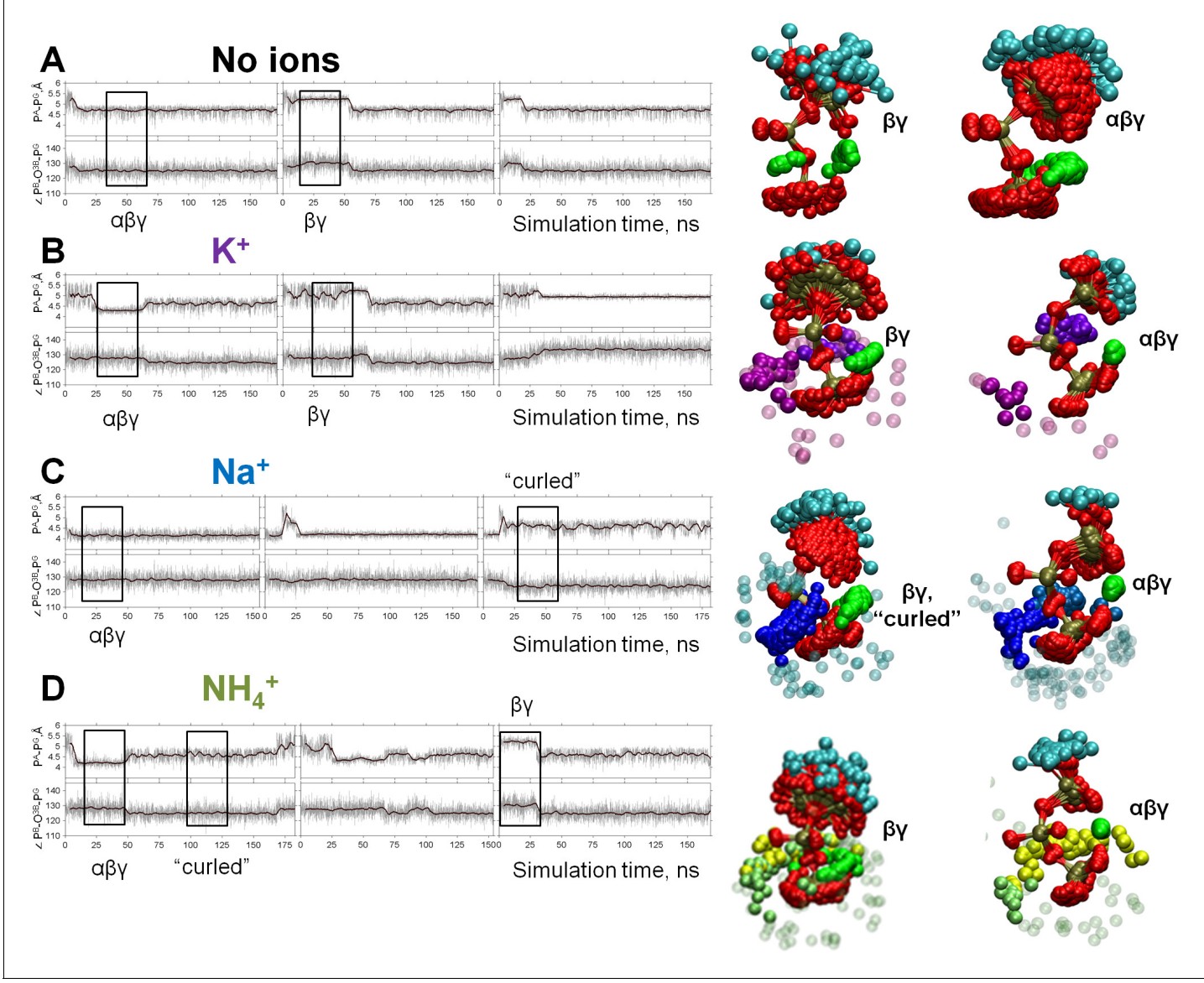

**Figure 4.** Dynamics of the phosphate chain of the Mg-ATP complex with and without monovalent cations. Each left panel shows the $P^A$-$P^G$ distance (upper trace) and the $P^B$-$O^{3B}$-$P^G$ angle (bottom trace) in the course of MD simulations. Thin gray lines show actual values measured from each frame of the MD simulation, the bold black lines show moving average with a 2-ps window. Black boxes indicate fragments of simulations chosen for the analyses of particular types of interaction between the $Mg^{2+}$ ion and the triphosphate chain; the respective conformations of Mg-ATP are shown on the right. The analysis was performed as shown in *Figure 2B*. The color scheme is as in *Figure 1*. (**A**) no added ions; (**B–D**) MD simulations in the presence of $K^+$, $Na^+$, and $NH_4^+$, respectively.

DOI: https://doi.org/10.7554/eLife.37373.009

The following figure supplements are available for figure 4:

**Figure supplement 1.** Coordination of the $Mg2^{2+}$ion by the oxygen atoms of the ATP phosphate chain during MD simulations.
DOI: https://doi.org/10.7554/eLife.37373.010
**Figure supplement 2.** Estimation of correlation times for the $P^A$-$P^G$ distances.
DOI: https://doi.org/10.7554/eLife.37373.011
**Figure supplement 3.** Estimation of correlation times for the $P^B$-O-$P^G$ angles.
DOI: https://doi.org/10.7554/eLife.37373.012

**Table 1.** Effects of monovalent cations on the shape of the triphosphate chain of the Mg-ATP complex in water, as inferred from the MD simulation data.

| | Conformation of the triphosphate chain of Mg-ATP[*] | | | | | |
| | βγ-coordination | | βγ-coordination, 'curled' phosphate chain | | αβγ-coordination | |
| Added cation | $P^A$-$P^G$ distance, Å | $P^B$-$O^{3B}$-$P^G$ angle | $P^A$-$P^G$ distance, Å | $P^B$-$O^{3B}$-$P^G$ angle | $P^A$-$P^G$ distance, Å | $P^B$-$O^{3B}$-$P^G$ angle |
|---|---|---|---|---|---|---|
| None | 5.46 ± 0.34 | 122.3 ± 3.5 | N/A | | 4.76 ± 0.18 | 124.9 ± 3.3 |
| $K^+$ | 4.91 ± 0.24 | 122.0 ± 3.3 | N/A | | 4.32 ± 0.24 | 128.0 ± 3.5 |
| $Na^+$ | 4.69 ± 0.22 | 122.9 ± 3.2 | 4.60 ± 0.22 | 124.0 ± 3.3 | 4.26 ± 0.37 | 127.7 ± 3.6 |
| $NH_4^+$ | 4.85 ± 0.22 | 122.3 ± 3.3 | 4.56 ± 0.21 | 124.6 ± 3.3 | 4.22 ± 0.16 | 127.8 ± 3. |

[*]The conformations of the Mg-ATP complex were determined as described in the text. Mean values and standard deviations of $P^A$-$P^G$ distance (in Å) and the $P^B$-$O^{3B}$-$P^G$ angle (in degrees) were measured over the respective parts of the simulations. Simulation periods corresponding to βγ and αβγ conformations were identified by tracking distances between $Mg^{2+}$ and non-bridging oxygen atoms of the phosphate chain (**Figure 4—figure supplement 1**); simulation periods corresponding to the 'curled' conformation were identified from $P^A$-$P^G$ distance tracks and visual inspection of the phosphate chain shape (**Figure 4**). Data for the αβγ coordination of the Mg-ATP complex and conformations with curled phosphate chain were calculated from simulations 1–4 in **Supplementary file 1C**; characterization of the βγ-coordination was based on simulations 5–8 in **Supplementary file 1C**, see **Supplementary file 1E** for further details.

DOI: https://doi.org/10.7554/eLife.37373.013

**Figure 5A**). Still, in the presence of $K^+$ ions, the occurrence of such transition state-like conformations was notably higher, while in simulations with $Na^+$ such conformations were completely absent.

Next, we compared the conformations of the phosphate chain of ATP (**Figure 5B**) and GTP (**Figure 5—figure supplement 1**) molecules that had βγ-coordinated $Mg^{2+}$ ion, as obtained in the series of short (20 ns) MD simulations (**Supplementary file 1C**), with the shapes of phosphate chains in the crystal structures of P-loop NTPases. As seen on the heat maps, in the absence of any $M^+$, the phosphate chain was remarkably elongated, displaying large $P^A$-$P^G$ distances that were not observed either in simulations with added cations or in crystal structures. The presence of $M^+$ ions led to the shortening of the $P^A$-$P^G$ distances. In the simulations with $Na^+$ ions, the ATP phosphate chain was more contracted than in the crystal structures of P-loop NTPases (**Figure 5B**, **Figure 5—figure supplement 1**). In contrast, in the MD simulations with $K^+$ and $NH_4^+$ ions, the phosphate chain shape matched almost exactly the conformations of the NTP analogs in the structures of P-loop NTPases. In MD simulations in the presence of $K^+$ and $NH_4^+$ ions, the distribution of the conformations of Mg-ATP complex spread over the areas of non-hydrolyzable NTP analogs and covered even transition state analogs (**Figure 5B**, **Figure 5—figure supplement 1**). Only the conformations of the transition state analogs with severely widened (>135°) $P^B$-$O^{3B}$-$P^G$ angle were not matched by the MD-derived conformations.

Altogether, **Figure 5** and **Figure 5—figure supplement 1** show that the conformational space of phosphate chain conformations, as seen in P-loop NTPases, overlapped much better with conformations seen in the MD simulations of Mg-ATP with $K^+$ and $NH_4^+$ ions than with conformations obtained with $Na^+$ ions.

## Cations in the active sites of P-loop NTPases

To further analyze the roles of $M^+$ ions in P-loop NTPases, we selected 10 crystal structures of P-loop GTPases and ATPases, representing different families of P-loop proteins. We have chosen mainly the structures with non-hydrolyzable NTP analogs and transition state analogs in complex with $Mg^{2+}$ ions, as these structures provide positions of all three phosphate groups. These structures were superposed by matching the coordinates of the P-loop regions against the structure of the $K^+$-dependent GTPase MnmE [PDB: 2GJ8] (**Scrima and Wittinghofer, 2006**), see **Figure 6**. Each structure was then inspected to determine the locations of the positively charged residues around the phosphate chain. **Figure 6** shows that the binding sites for $M^+$ ions observed in the MD simulations (**Figure 6A**) were exactly those occupied by positively charged groups in the structures of P-loop NTPases (**Figure 6B,C**). The binding site between the β- and γ-phosphates (the BG site) is always occupied by the amino group of the conserved P-loop lysine residue, whereas the binding site

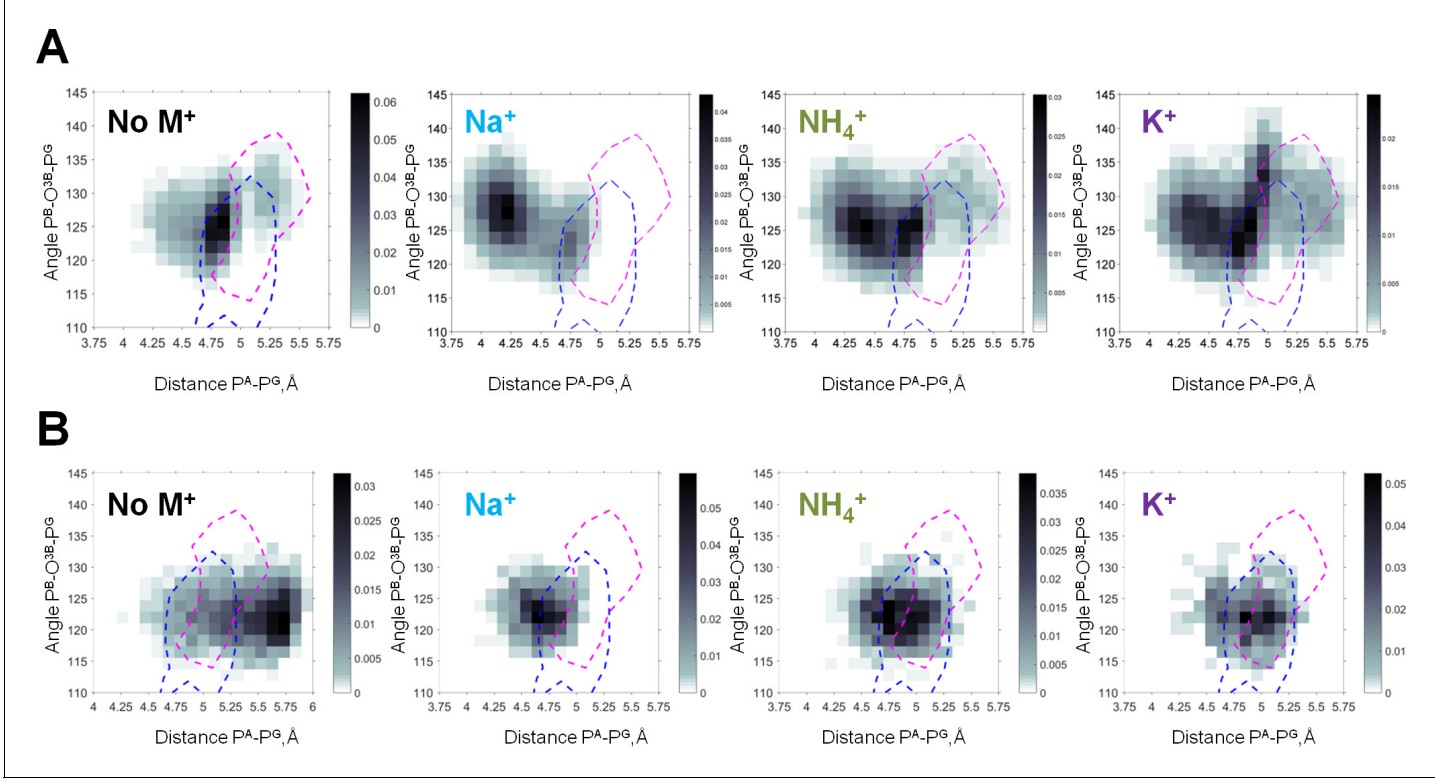

**Figure 5.** Heat maps of the Mg-ATP phosphate chain conformations distribution characterized by the $P^A$-$P^G$ distances (X-axis) and $P^B$-$O^{3B}$-$P^G$ angles (Y-axis). Heat maps for systems with monovalent cations include only conformations of Mg-ATP complexes with at least one cation present within 4 Å radius. The color intensity is proportional to the probability (normalized frequency) of the respective conformation. Magenta dashed lines outline the areas corresponding to the conformations of transition state analogs; blue dashed lines outline the areas corresponding to the conformations of the non-hydrolyzable analogs, calculated from crystal structures of P-loop NTPases (*Figure 5—figure supplement 2*). (**A**) Data from the 3 × 170 ns simulations (no. 1–4 in *Supplementary file 1C*). (**B**) Data from 4 × 20 ns simulations of Mg-ATP in βγ conformations (no. 5–8 in *Supplementary file 1C*).
DOI: https://doi.org/10.7554/eLife.37373.014

The following figure supplements are available for figure 5:

**Figure supplement 1.** Heat maps of the Mg-GTP phosphate chain conformations distribution characterized by the $P^A$-$P^G$ distances (X-axis) and $P^B$-$O^{3B}$-$P^G$ angles (Y-axis).
DOI: https://doi.org/10.7554/eLife.37373.015

**Figure supplement 2.** Phosphate chain shape of ATP and GTP analogs in the X-ray structures of P-loop NTPases.
DOI: https://doi.org/10.7554/eLife.37373.016

between the α- and γ-phosphates (the AG site) could be occupied, in the crystal structures, by either a $K^+$ or $Na^+$ ion (*Figure 6B*), or an amino group of an activating lysine residue, or the guanidinium group of arginine (*Figure 6C*), or a water molecule (see below).

In all P-loop NTPases, the phosphate chain is seen in the extended conformation similar to that observed in the presence $K^+$ and $NH_4^+$ but not $Na^+$ ions (*Figure 5*, *Figure 5—figure supplement 1*). Such an extended conformation is known to be stabilized by numerous interactions of all three phosphate groups with the residues of the P-loop motif, see (*Wittinghofer and Vetter, 2011*).

*Table 2* summarizes the activation mechanisms for those classes of P-loop NTPases that contain both $M^+$-activated and Arg/Lys-activated enzymes. Across different families of P-loop NTPases, different activation mechanisms have been described, usually involving interactions with other proteins, domains of the same protein, or RNA/DNA molecules, and resulting in the insertion of a positive charge - a monovalent cation or an Arg/Lys finger - into the catalytic site (*Bos et al., 2007*; *Scrima and Wittinghofer, 2006*; *Ash et al., 2011*; *Meyer et al., 2009*; *Böhme et al., 2010*; *Abrahams et al., 1994*; *Goitre et al., 2014*; *Komoriya et al., 2012*; *Vetter and Wittinghofer, 1999*). The catalytic roles of Arg/Lys residues in the AG sites of various classes of P-loop NTPases

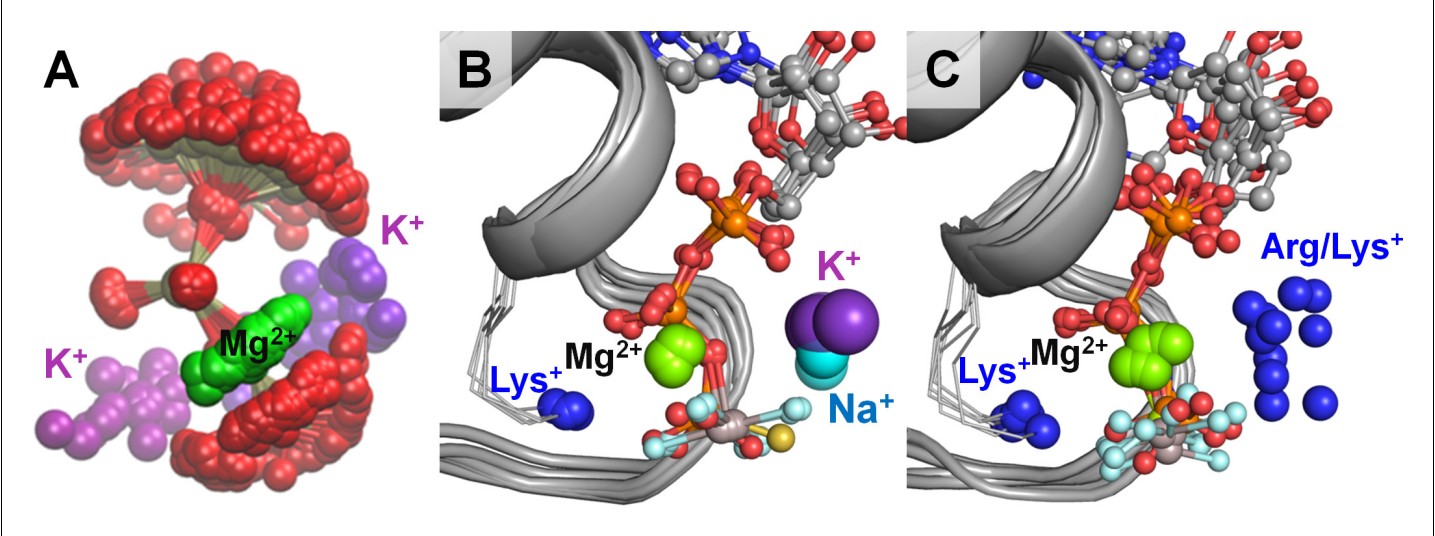

**Figure 6.** Location of positive charges around the phosphate chain of Mg-NTP complexes in solution and in protein structures. The color scheme is as in *Figure 1*; dark blue spheres indicate positions of positively charged side-chain nitrogen atoms of Lys and Arg residues, P-loop regions are shown as cartoons in grey. (A) Superposition of phosphate chain conformations observed in MD simulations with $K^+$ ions. Only conformations with $\beta\gamma$ coordination of $Mg^{2+}$ are shown. (B) Superposition of P-loop regions of crystal structures of cation-dependent P-loop NTPases: GTPase MnmE [PDB: 2GJ8], Fe transporter FeoB [PDB: 3SS8], dynamin-like protein [PDB: 2X2E], and translation factor eIF-B5 [PDB: 4TMZ], see *Table 3* for details. (C) Superposition of P-loop regions of crystal structures of cation-independent P-loop NTPases: Ras/RasGAP complex [PDB: 1WQ1], septin [PDB: 3FTQ], atlastin [PDB: 4IDQ], $G_{\alpha 12}$ protein [PDB: 1ZCA], DNA polymerase III subunit $\tau$ [PDB: 3GLF], $F_1$-ATPase [PDB: 2JDI].
DOI: https://doi.org/10.7554/eLife.37373.017

The following figure supplements are available for figure 6:

**Figure supplement 1.** Active sites of P-loop NTPases with established $K^+$-dependent activity (see *Supplementary file 1A* for the full list and references).
DOI: https://doi.org/10.7554/eLife.37373.018

**Figure supplement 2.** Activation of the MnmE GTPase upon dimerization.
DOI: https://doi.org/10.7554/eLife.37373.019

**Figure supplement 3.** Activation of the GTPase Era upon RNA binding.
DOI: https://doi.org/10.7554/eLife.37373.020

**Figure supplement 4.** Positively charged moieties in the active site of RecA-like recombinases.
DOI: https://doi.org/10.7554/eLife.37373.021

are discussed elsewhere (*Shalaeva et al., 2018*). Here, we focus on the structures of P-loop NTPases that are dependent upon $M^+$ ions.

We have manually inspected the available structures of known $K^+$-dependent P-loop NTPases (*Supplementary file 1A*), checked for $M^+$ ions bound near the NTP phosphate chain, and compared the structures of $K^+$- and $Na^+$-bound NTP analogs in crystal structures of P-loop proteins with the structures of the $Mg^{2+}$-ATP-$2K^+$ and $Mg^{2+}$-ATP-$2Na^+$ complexes obtained from the MD simulations. In total, we were able to identify and analyze 17 structures of cation-dependent P-loop NTPases in complex with NTP analogs and $K^+$, $Na^+$, or $NH_4^+$ ions bound in the active site (*Table 3*). For each such structure, we checked the shape of the phosphate chain and the coordination sphere of the cation in the AG site. In all these structures, the distances between $P^A$ and $P^G$ atoms (or between the corresponding mimicking atoms) were in the range of 4.9–5.3 Å for the non-hydrolyzable analogs and 5.3–5.6 Å for transition state analogs (*Table 3*). These values are similar to the $P^A$-$P^G$ distances observed in MD simulations of the Mg-ATP complex in the presence of $K^+$ ions (*Figures 4* and *5* and *Table 1*).

The majority of $K^+$-activated NTPases, as well as the unique family of the $Na^+$-adapted dynamin-related GTPases, belong to the TRAFAC class of P-loop NTPases (*Leipe et al., 2002*), where the binding of the $M^+$ ion is assisted by the so-called K-loop (*Ash et al., 2012*). This loop goes over the nucleotide binding site and provides two backbone carbonyl groups as additional ligands to the $M^+$ coordination sphere (purple cartoon and sticks in *Figure 1B,C*). To our surprise, very few structures

**Table 2.** Activation mechanisms within the classes of P-loop NTPases that contain both cation-dependent and cation-independent enzymes.

| Superfamily | Family | Activating charge | Activation mechanism |
|---|---|---|---|
| **Kinase-GTPase division, TRAFAC class** | | | |
| Classic translation factor GTPases | EF-G/EF-2 | $K^+$ | Functional interaction with ribosomal RNA/other protein(s)/other domain(s) of the same protein (*Hwang and Inouye, 2001*;*Moreau et al., 2008*; *Tomar et al., 2011*; *Achila et al., 2012*; *Fasano et al., 1982*; *Ebel et al., 1992*; *Dubnoff and Maitra, 1972*; *Kuhle and Ficner, 2014*; *Manikas et al., 2016*; *Daigle and Brown, 2004*; *Foucher et al., 2012*; *Rafay et al., 2012*; *Pérez-Arellano et al., 2013*; *Villarroya et al., 2008*) |
| | EF-Tu/EF-1A | $K^+$ | |
| | EIF2G | $K^+$ | |
| | ERF3 | $K^+$ | |
| | IF-2 | $K^+$ | |
| | LepA | $K^+$ | |
| OBG-HflX-like GTPases | HflX | $K^+$ | |
| | OBG | $K^+$ | |
| | NOG | $K^+$ | |
| | YchF/OLA1 | $K^+$ | |
| YlqF/YawG GTPases | NOG2 | $K^+$ | |
| | RsgA | $K^+$ | |
| TrmE-Era-EngA-EngB-Septin-like GTPases | EngA (Der) | $K^+$ | |
| | EngB | $K^+$ | |
| | Era | $K^+$ | |
| | FeoB | $K^+$ | Dimerization (e.g. mRNA-associated in the case of MnmE) (*Chappie et al., 2010*; *Koenig et al., 2008*; *Gasper et al., 2009*) |
| | MnmE | $K^+$ | |
| | Septin | Arg finger | |
| | Toc34-like | Arg finger | |
| Dynamin-like GTPases | hGBP | Arg finger | |
| | | Dynamin | |
| **$K^+$/$Na^+$** | | | |
| Extended Ras | Ras family | Arg finger | Interaction with a specialized activating protein or domain (*Bos et al., 2007*; *Cherfils and Zeghouf, 2013*) |
| | Gα | subunits | |
| **Arg finger** | | | |
| Myosin/kinesin | Myosin | Arg finger | |
| | Kinesin | Arg finger | |
| **ASCE division, RecA/F1-like class** | | | |
| DNA-repair and recombination ATPases | RecA | Lys finger | DNA/RNA-dependent oligomerization (*Chen et al., 2008*) |
| | RadA | $K^+$ | |

*Table 2 continued on next page*

*Table 2 continued*

| Superfamily | Family | Activating charge | Activation mechanism |
|---|---|---|---|
| Rho helicases | Rho | Arg finger | Interaction with the neighboring subunit within a conformationally coupled hexamer (**Komoriya et al., 2012**; **Walker, 1998**; **Senior et al., 2002**; **Skordalakes and Berger, 2006**) |
| T3SS ATPases | YscN | Arg finger | |
| | FliI | Arg finger | |
| F-/V-type ATPases | V-type A | Arg finger | |
| | F-type β | | |
| | V-type B | | |
| | F-type α | | |

DOI: https://doi.org/10.7554/eLife.37373.022

**Table 3.** Monovalent cation binding in crystal structures of P-loop NTPases.

| Protein | PDB entry | Bound NTP analog | Occupation of the AG site | | | Phosphate chain shape | |
|---|---|---|---|---|---|---|---|
| | | | Cation | Distance to the closest O atom of $P^A$, Å[*] | Distance to the closest O atom of $P^G$, Å[*,†] | $P^A$-$P^G$ distance, Å[*] | $P^B$-$O^{3B}$-$P^G$ angle, degrees[†] |
| **TRAFAC class NTPases** | | | | | | | |
| GTPase MnmE(TrmE) | 2gj8 | GDP AlF$_4^-$ | K$^+$ | 2.8 | 2.6 | 5.4 | 136.3 |
| | 2gja | GDP AlF$_4^-$ | NH$_4^+$ | 2.9 | 2.5 | 5.4 | 136.9 |
| | 2gj9 | GDP AlF$_4^-$ | Rb$^+$ | 2.9 | 2.8 | 5.5 | 131.6 |
| GTPase FeoB | 3ss8 | GDP AlF$_4^-$ | K$^+$ | 2.8 | 2.6 | 5.4 | 144.9 |
| Dynamin-like proteins | 2x2e | GDP AlF$_4^-$ | Na$^+$ | 4.0 | 2.5 | 5.3 | 131.2 |
| | 2x2f | GDP AlF$_4^-$ | Na$^+$ | 4.1 | 2.6 | 5.3 | 133.6 |
| | 3w6p | GDP AlF$_4^-$ | Na$^+$ | 4 | 2.4 | 5.5 | 135.3 |
| | 3t34 | GDP AlF$_4^-$ | Na$^+$ | 3.8 | 2.4 | 5.6 | 149.3 |
| GTPase Era | 3r9w | GNP | H$_2$O[‡] | 3 | 3.4 | 5.1 | 129.2 |
| Eukaryotic translation initiation factor eIF5B | 4ncn | GTP | Na$^+$ | 2.4 | 2.4 | 5.0 | 126.6 |
| | 4tmv | GSP | Na$^+$ | 2.4 | 2.8 (S)[§] | 4.9 | 126.3 |
| | 4tmw | GTP | Na$^+$ | 2.4 | 2.4 | 4.9 | 125.9 |
| | 4tmz | GSP | K$^+$ | 2.7 | 3.3 (S)[§] | 4.9 | 122.1 |
| **RecA/F1-like class NTPases** | | | | | | | |
| DNA recombinase RadA | 3ew9 | ANP | K$^+$ | 6.2 | 3.3 | 5.1 | 124.5 |
| | 2f1h | ANP | K$^+$ | 6.6 | 3.5 | 5.3 | 125.3 |
| | 2fpm | ANP | K$^+$ | 5.9 | 2.6 | 5.1 | 124.2 |
| | 1xu4 | ANP | K$^+$ | 6.1 | 2.7 | 5.2 | 125.0 |

[*]The values were measured directly in the respective protein structures displayed in PyMOL.

[†] If the γ-phosphate was replaced by an AlF$_4^-$ complex, the distance was measured to the closest F atom

[‡] While GTPase Era has been shown to be K$^+$-dependent (**Rafay et al., 2012**; **Meier et al., 2000**), the crystallization solution contained no K$^+$, only Na$^+$, so that the likely cation-binding site is occupied by a water molecule, which forms hydrogen bonds with K$^+$ ligands.

[§] Non-hydrolyzable GTP analog GDP-monothiophosphate (GSP) contains a sulfur atom in the place of the $O^{1G}$ atom of γ-phosphate; this atom in involved in coordination of monovalent cations in respective structures.

DOI: https://doi.org/10.7554/eLife.37373.023

of K$^+$-dependent GTPases of the TRAFAC class contained K$^+$ ions in their AG sites (cf *Supplementary file 1A* and *Table 3*). Furthermore, in most cases, the K$^+$ loops were either unresolved or distorted (*Figure 6—figure supplement 1*). Separate crystal structures with and without activating K$^+$ ion were available only for the tRNA modification GTPase MnmE, see *Table 3* and *Figure 6—figure supplement 2*. It is believed that during the catalytic turnover, two MnmE proteins undergo conformational changes to allow dimerization of their P-loop GTPase domains (G-domains) resulting in their mutual activation (*Meyer et al., 2009*; *Böhme et al., 2010*). We have compared the two structures of MnmE GTPase to further clarify their K$^+$-binding determinants. In the crystallized full-length MnmE dimer, only the N-terminal domains of the two proteins interact, forming a central hinge, whereas the large helical domains and G-domains are located on the opposite sides from the central hinge (PDB: 3GEI, *Figure 6—figure supplement 2*). In such an arrangement, the distance between the active sites of the G-domains (with non-hydrolyzable GTP analogs bound) is about 20 Å (*Scrima and Wittinghofer, 2006*; *Sehorn et al., 2004*). The K-loops, responsible for cation binding, are not resolved, and no K$^+$ binding is observed. In the crystal structures of the isolated G-domains of MnmE in complex with the transition state analog GDP-AlF$_4^-$, which are dimerized via their K-loop (Switch I) regions (as defined in *Figure 1*), the K-loops and M$^+$ cations are resolved (PDB: 2GJ8, *Figure 6—figure supplement 2*). The disordered K-loop in the inactive state of MnmE and the stabilized K-loop in the active state of the protein indicate that the activity of the enzyme could be controlled via formation of a full-fledged K$^+$-binding site upon dimerization.

In one of the structures of the K$^+$-dependent GTPase Era, which was crystallized in the absence of K$^+$ ions (PDB: 3R9W, [*Tu et al., 2011*]), the potential K$^+$ binding site contains a water molecule (id 624) that is 2.9–3.4 Å away from six potential K$^+$ ion ligands. Owing to the presence of a full-fledged K$^+$-binding site, we included this structure in *Table 3* (see also *Figure 6—figure supplement 3*).

Outside of the TRAFAC class, only a few cases of K$^+$-dependent P-loop NTPases are known, all among RecA-like recombinases (*Tables 2* and *3* and *Supplementary file 1A*). Along with rotary ATPases, these proteins are assigned to the RecA/F1-like class of the ASCE (Additional Strand, Catalytic E) division, as they bear an additional strand between the Walker A and Walker B motifs and have a conserved Glu residue in the catalytic site (*Leipe et al., 2002*). Consequently, RecA-like recombinases are dramatically different from the TRAFAC class proteins and lack such characteristic structural motifs as Switch I/K-loop and Switch II. Crystal structure of the K$^+$-dependent recombinase RadA [PDB: 3EW9] (*Li et al., 2009*) shows two binding sites for K$^+$ ions (*Figure 6—figure supplement 4*). One of these binding sites corresponds roughly to the AG site, although the cation is shifted towards γ-phosphate and away from α-phosphate. The second cation is bound between the γ-phosphate and the catalytic Glu residue, in the position that corresponds to the low-occupancy G-site observed in our MD simulations in water (*Figure 2A*).

## Molecular dynamics simulations of MnmE GTPase

The GTPase MnmE is the only K$^+$-dependent-NTPase for which both K$^+$- bound (PDB: 2GJ8, resolution 1.7 Å, source: *E. coli*) and K$^+$-free (PDB: 3GEI, resolution 3.4 Å, source: *Chlorobium tepidum*) X-ray structures are available (*Figure 6—figure supplement 2*). To clarify whether the binding of the K$^+$ ion could affect the shape of the phosphate chain of Mg-GTP in the active site of MnmE, we performed MD simulations of this protein in its active and inactive states (*Figure 7*). To model the active state, we took the X-ray structure of the dimeric GTPase domain (G-domain) from *E. coli* with a K$^+$ ion bound in each of the two active sites (PDB: 2GJ8)) and replaced the transition state analogs GDP-AlF$_4^-$ in the two active sites by GTP molecules (hereafter the 2GJ8$_K$ system). The inactive, K$^+$ free state was modeled by two systems. One inactive system was the monomer from the same 2GJ8 crystal structure where the GDP-AlF$_4^-$ complex was replaced by a GTP molecule and the K$^+$ ion was replaced by a water molecule (hereafter the 2GJ8$_W$ system). The monomeric form was chosen because no dimerization takes place in the absence of K$^+$ ions (*Meyer et al., 2009*). Another modeled inactive system was the monomeric G-domain of MnmE from the full-length structure of the MnmE from *Chlorobium tepidum* (PDB 3GEI), where the non-hydrolysable GTP analog guanosine 5′-imidotriphosphate (GppNHp) was replaced by a GTP molecule and the non-resolved K-loop (see above) was reconstructed (hereafter the 3GEI system). These three systems were placed in water boxes with KCl and simulated for 100 ns each (*Supplementary file 1C*). During the simulation of the inactive 3GEI system, the displacement of the reconstructed K-loop led to the loss of the coordinating bond between the Mg$^{2+}$ ion and Thr268 (which corresponds to Thr251 in the *E. coli* structure,

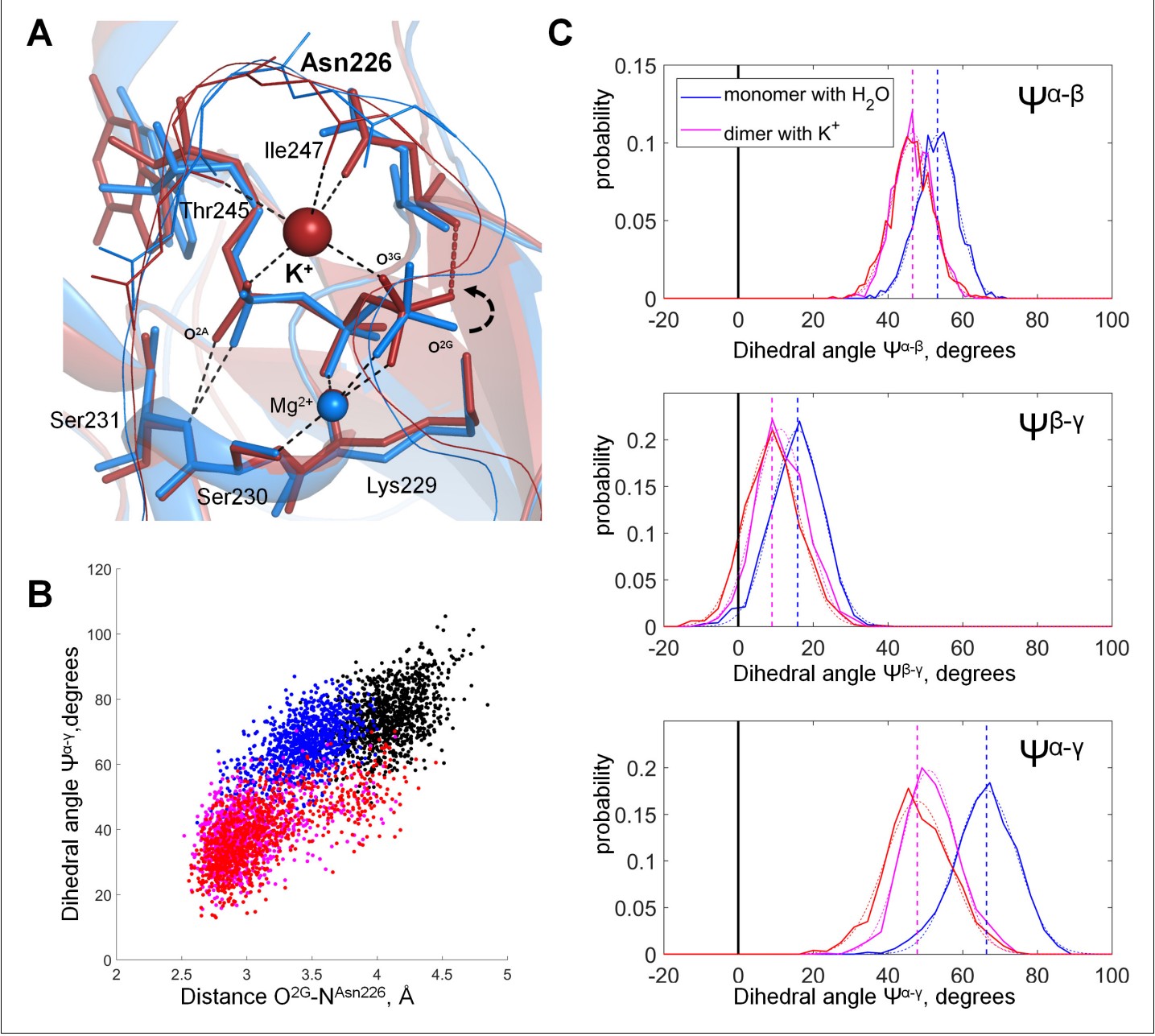

**Figure 7.** Molecular dynamics of MnmE GTPase. (**A**) Superposition of the GTP-binding sites of the inactive, monomeric G-domain of MnmE (the 2GJ8$_W$ system, blue) and the active K$^+$-bound dimer of G-domains (the 2GJ8$_K$ system, red); representative structures were sampled from the last 10 ns of 100 ns simulations. The protein backbones are shown as cartoons; GTP and surrounding amino acid residues are shown as sticks; Mg$^{2+}$ and K$^+$ ions are shown as spheres. Black dashed lines indicate hydrogen bonds and coordination bonds for cations that are present in both structures; the red dashed line indicates the H-bond that is present only in the K$^+$-containing 2GJ8$_K$ system. (**B**) Conformational space of GTP in different states of MnmE GTPase. Scatter plot of the $\Psi^{\alpha-\gamma}$ dihedral angle (Y-axis) against the length of the hydrogen bond between the O$^{2G}$ atom and the backbone nitrogen of Asn226 (X-axis) as sampled from the MD simulations of three systems: (1) red/orange, active dimer of G-domains with K$^+$ ions bound (the 2GJ8$_K$ system, red and orange for individual monomers); (2) blue, monomeric G-domain of MnmE with the K$^+$ ion replaced by a water molecule (the 2GJ8$_W$ system); and (3) black, inactive monomer G-domain of MnmE without a full-fledged K-loop (the 3GEI system). (**C**) Distribution histograms for dihedral angles between the phosphate groups in GTP, calculated from MD simulations of the dimeric G-domain of MnmE with bound K$^+$ ions (the 2GJ8$_K$ system, red and magenta colors represent individual monomers in the dimer) and the monomeric G-domain with the K$^+$ ion replaced by a water molecule (the 2GJ8$_W$ system, blue). Normalized histograms of dihedral angle distribution (solid lines) were calculated from MD trajectories and fitted with normal distribution function (dotted lines). Vertical lines indicate the centroid values of the fits by Gaussian function. Black vertical lines indicate $\Psi$ = 0°, which corresponds to the fully eclipsed conformation, while $\Psi$ =± 60° corresponds to the fully staggered conformation.

DOI: https://doi.org/10.7554/eLife.37373.024

*Figure 7 continued on next page*

*Figure 7 continued*

The following figure supplements are available for figure 7:

**Figure supplement 1.** Hydrogen bonds lengths during MD simulations of the G-domain of MnmE.
DOI: https://doi.org/10.7554/eLife.37373.025
**Figure supplement 2.** The distance between $O^{2A}$ and $O^{3G}$ in GTP bound to MnmE as inferred from MD simulations.
DOI: https://doi.org/10.7554/eLife.37373.026

PDB 2GJ8). The bidentate βγ coordination of $Mg^{2+}$ got distorted and the phosphate chain fluctuated between the βγ- and αβγ-coordination states. In spite of these fluctuations, the α-phosphate-binding H-bond between the backbone nitrogen of Thr248 (which corresponds to Ser231 in the *E. coli* structure) and the $O^{1A}$ atom remained as stable throughout the MD simulation (*Figure 7—figure supplement 1A*) as the corresponding H-bond between the backbone nitrogen of Ser231 and the $O^{1A}$ atom in the $2GJ8_K$ and $2GJ8_W$ systems (*Figure 7—figure supplement 1B,C*). In spite of the absence of $K^+$ ions in the $2GJ8_W$ system, the K-loop stayed in place during the MD simulation and prevented the destabilization of the phosphate chain (*Figure 7—figure supplement 1A*).

Introduction of the $K^+$ ion between $O^{2A}$ and $O^{3G}$ atoms led to the shortening of the distance between these atoms (*Figure 7A*, *Figure 7—figure supplement 2*). In the MD simulation of the inactive $2GJ8_W$ system, the average distance between these two oxygen atoms was 5.3 Å, whereas in the active $2GJ8_K$ dimer, the average distance between the oxygen atoms decreased to 4.7 Å (*Figure 7—figure supplement 2*). By pulling the $O^{3G}$ atom closer to $O^{2A}$, the $K^+$ ion twisted the γ-phosphate, so that its $O^{2G}$ atom formed a new H-bond with the backbone nitrogen of Asn226 (*Figure 7A,B*). It is noteworthy that the side chain of this residue is directly coordinating the $K^+$ ion (*Figure 7A*). This H-bond between the $O^{2G}$ atom and Asn226 was not seen in the inactive 3GEI and $2GJ8_W$ systems (*Figure 7B*, *Figure 7—figure supplement 1A,B*).

To specify the conformational changes of the phosphate chain in response to the insertion of the $K^+$ ion in the AG site, we measured the values of dihedral angles between phosphate groups (as defined in *Figure 3C*), namely $\Psi^{\alpha-\beta} = \angle O^{2A}\text{-}P^A\text{-}P^B\text{-}O^{2B}$, $\Psi^{\beta-\gamma} = \angle O^{1B}\text{-}P^B\text{-}P^G\text{-}O^{1G}$, and $\Psi^{\alpha-\gamma} = \angle O^{1A}\text{-}P^A\text{-}P^G\text{-}O^{3G}$ (*Figure 7C*). Since in the 3GEI system the βγ coordination of $Mg^{2+}$ by GTP was not retained, further we compare only the $2GJ8_K$ and $2GJ8_W$ systems. In our simulations, the α-phosphate oxygen atoms remained close to the energetically favorable staggered conformation relative to the β-phosphate oxygen atoms in both MD simulation systems; upon insertion of the $K^+$ ion, the $\Psi^{\alpha-\beta}$ angle decreased only slightly from 53° to 47° (*Figure 7C*). Owing to the coordination of β- and γ-phosphates by the $Mg^{2+}$ ion and Lys229, the dihedral angle between β- and γ-phosphates was close to the eclipsed value of zero degrees in both MD simulation systems (6° in the $2GJ8_K$ system and 16° in the $2GJ8_W$ system, *Figure 7A,C*). The dihedral angle between α- and γ-phosphates was close to that for the staggered conformation in the $2GJ8_W$ system (~66°, *Figure 7A,C*). However, in the MD simulation of the $2GJ8_K$ system, the γ-phosphate rotated by ca. 30° towards the eclipsed conformation relative to the α-phosphate yielding $\Psi^{\alpha-\gamma}=38°$ (*Figure 7A,C*). Thus, the insertion of the $K^+$ ion between the $O^{2A}$ and $O^{3G}$ atoms brought the phosphate chain into a state where all three phosphate groups were more eclipsed (*Figure 7A,C*). The eclipsed conformation was achieved largely due to the rotation of the γ-phosphate group, accompanied by minor movements of α- and β-phosphates (*Figure 7A,C*). The γ-phosphate rotated from a staggered conformation towards the near-eclipsed conformation with respect to α and β-phosphate (*Figure 7A,C*). This rotation was accompanied by formation of a new H-bond with Asn226 (*Figure 7*, *Figure 7—figure supplement 1*).

## Discussion

### Activation of P-loop NTPases by monovalent cations

The hydrolysis of NTPs is a key reaction in biochemistry. The large amount of free energy that is released upon the hydrolysis of NTPs results from the repulsion between the negatively charged phosphate groups. At the same time, the cumulative negative charge of these groups repels the attacking nucleophilic groups (usually the $OH^-$ ions), securing the stability of the molecule in the absence of NTPases (*Kamerlin et al., 2013*; *Westheimer, 1987*). So far, computational studies of

the mechanisms of NTP hydrolysis in water used such model systems as methyl triphosphate molecule with and without $Mg^{2+}$, Mg-ATP complex, and Mg-GTP complex, see for example (*Akola and Jones, 2003*; *Grigorenko et al., 2006*; *Harrison and Schulten, 2012*; *Liao et al., 2004*; *Simonson and Satpati, 2013*). These modeled systems, however, contained no monovalent cations.

The current views on the mechanisms of NTP hydrolysis (*Kamerlin et al., 2013*; *Jin et al., 2017a*; *Jin et al., 2017b*; *Akola and Jones, 2003*; *Grigorenko et al., 2006*; *Harrison and Schulten, 2012*; *Westheimer, 1987*) posit that the electrophilic γ-phosphorus atom is attacked by a hydroxyl group, derived from a pre-polarized water molecule. To facilitate access of the negatively charged hydroxyl to the phosphate chain, several positive charges are needed to compensate the four negative charges of the phosphate groups and, additionally, the transient negative charge of the leaving NDP group, see (*Kamerlin et al., 2013*; *Jin et al., 2017a*) for reviews. Catalysis by known NTPases utilizes at least four positive charges: either two divalent cations (as in DNA and RNA polymerases and many nucleases and transposases (*Mildvan, 1987*; *Yang et al., 2006*)) or a divalent cation (usually $Mg^{2+}$) and two single positive charges. In P-loop NTPases, these single positive charges are provided by (i) the conserved Lys residue of the P-loop and (ii) the activating $M^+$ ion or Lys/Arg residue.

Here, we showed that these positively charged, catalytic moieties, in the crystal structures of diverse P-loop NTPases, occupy exactly those sites that were occupied by $M^+$ ions upon MD simulations of Mg-NTP complexes in water (*Figure 6*). The $M^+$ ion-binding sites between the β and γ phosphates (the BG sites) are always occupied by the $NH_3^+$ group of the highly conserved P-loop lysine residue. The AG sites are usually taken by the activating moiety - either by a positively charged arginine or lysine residue (*Figures 1A* and *6C*) or by a $M^+$ ion (*Figures 1B, C*, *6B* and *7*) (*Scrima and Wittinghofer, 2006*; *Ash et al., 2012*; *Meyer et al., 2009*; *Böhme et al., 2010*). Even in the absence of enzymes, $K^+$ and $NH_4^+$ ions, by occupying specific well-defined sites, yielded a specific extended conformation of the triphosphate chain. This conformation was similar to that in the active sites of P-loop NTPases (*Figure 6*). In the enzyme active sites, the strictly conserved extended conformation of NTPs or NTP analogs is maintained mostly by their interactions with the side chains and backbone atoms of the P-loop motif (*Wittinghofer and Vetter, 2011*). Specifically, in case of the MnmE GTPase of *E.coli*, the $O^{1A}$ atom forms a H-bond with the backbone nitrogen of Ser231 (*Figure 7—figure supplement 1*), whereas $O^{1B}$ and $O^{2G}$ atoms are coordinated by the amino group of Lys229 (*Figures 1B* and *7*). The $O^{2B}$ and $O^{1G}$ atoms coordinate the $Mg^{2+}$ ion, together with oxygen atoms of Ser230 of the P-loop motif, Thr251 of the K-loop, and two water molecules. The oxygen atoms of the phosphate chain additionally formed several H-bonds with the backbone nitrogens of the P-loop residues: $O^{1B}$ with Gly228 and Ala227, and $O^{3B}$ with Asn226 (*Table 1*, *Figures 1B* and *7*, *Figure 7—figure supplement 1*). This set of electrostatically compensating bonds appears to firmly fix the phosphate chain in the extended conformation (*Figures 1*, *6* and *7*).

The stabilization of an NTP molecule in an extended conformation by the P-loop dramatically increases the rate of hydrolysis even in the absence of an activating moiety. For instance, in Ras-like GTPases, binding of GTP to the P-loop accelerates the rate of hydrolysis by five orders of magnitude (*Kötting and Gerwert, 2004*; *Shutes and Der, 2006*). Hence, the P-loop-bound, extended conformation of the phosphate chain (*Figure 6*) appears to be already catalytically prone. Delbaere and coauthors noted that, in a bound NTP molecule, β- and γ-phosphates are in an eclipsed state owing to their interactions with the $Mg^{2+}$ ion and conserved Lys residue of the P-loop. In such a state, the β- and γ-phosphates repel each other, which could explain the higher hydrolysis rate (*Delbaere et al., 2004*; *Matte et al., 1998*).

Building on the structure comparison, we suggest that the extended conformation, which is typical for Mg-NTP complexes in water in the presence of large monovalent cations (*Table 1*, *Figures 5* and *6*, *Figure 5—figure supplement 1*), is more prone to hydrolysis than the more compressed conformation that we observed in the presence of the smaller $Na^+$ ions (*Figures 4* and *5*, *Figure 5—figure supplement 1*). This suggestion could explain why larger ions, such as $K^+$ and $Rb^+$, were more efficient than the smaller $Na^+$ and $Li^+$ ions in accelerating transphosphorylation even in the absence of enzymes (*Lowenstein, 1960*), see *Supplementary file 1B*.

Insertion of an activating moiety in the active site of P-loop NTPases accelerates the hydrolysis by additional several orders of magnitude (*Wittinghofer, 2006*; *Kamerlin et al., 2013*). There is no common view on the mechanism of this acceleration, see (*Wittinghofer and Vetter, 2011*; *Kamerlin et al., 2013*; *Gerwert et al., 2017*; *Jin et al., 2017a*; *Jin et al., 2017b*) for recent reviews.

Our MD simulations showed that binding of a $K^+$ ion in the AG site was accompanied by the transition of the triphosphate chain into an almost eclipsed conformation both in the case of Mg-ATP in water (*Figure 3*) and in MnmE GTPase (*Figure 7*). For the Ras GTPase and heterotrimeric G proteins, Gerwert and colleagues proposed that the Arg finger (an analog of $K^+$ ion) brings the phosphate chain into a fully eclipsed, catalytically productive conformation by rotating the α-phosphate group relatively to the β- and γ-phosphates (*Rudack et al., 2012*; *Mann et al., 2016*; *Gerwert et al., 2017*). Their suggestion was based on the results of QM/MM simulations of methyltriphosphate in water (*Rudack et al., 2012*), where the rotation of the α-phosphate was not restricted. In the case of Mg-ATP in water, we have also observed that binding of the second cation in the AG site, in the position of the activating Arg residue of Ras GTPase, caused a major rotation of the α-phosphate that, together with a minor rotation of the more constrained β- and γ-phosphates, yielded a fully eclipsed conformation of the phosphate chain (*Figure 3*). However, in the NTP molecule that is bound by the P-loop, the rotation of α-phosphate is prevented by interactions of the whole NTP molecule with the enzyme. For instance, in MnmE, α-phosphate forms a stable H-bond with the backbone amide group (of Ser231 in *E. coli* and Thr 248 in *C. tepidum*, *Figure 7—figure supplement 1*), this bond persisted even in MD simulations of the inactive 3GEI system that lacked a full-fledged K-loop (*Figure 7—figure supplement 1*). Still, even the breakage of this H-bond would not permit rotation of the α-phosphate because the moieties from the both sides of α-phosphate (the nucleobase-ribose and Mg-bound β- and γ-phosphates, respectively) are tightly fixed owing to multiple interactions with the protein (*Figures 1* and *7*). Therefore, the rotation around the bridging bonds of $P^A$ is impeded; the neighboring moieties cannot rotate within the active site, especially when the P-loop domain interacts with its activator.

Our MD simulations of the MnmE GTPase showed that, because of the extended conformation of the triphosphate chain imposed by the P-loop, the $K^+$ ion could not connect $O^{2A}$ and $O^{3G}$ atoms of α- and γ-phosphates without bringing them into an eclipsed arrangement relative to each other (see *Figure 7*, *Figure 7—figure supplement 2*). The near-eclipsed conformation of the phosphate chain was achieved mostly via rotation of the γ-phosphate (*Figure 7*). In contrast to the α-phosphate, the γ-phosphate, being the end group, could rotate even within the complex between the P-loop domain and its activator. The rotation of γ-phosphate is constrained only by its H-bonding with the conserved Lys229 residue of the P-loop, the backbone amide groups of flexible Gly-rich loop regions, and the coordination bond with the $Mg^{2+}$ ion. Owing to the ability of the Lys side chain to stretch out (*Cherepanov and Mulkidjanian, 2001*), this network of bonds around γ-phosphate appears to be flexible/elastic enough to permit a pronounced rotation of γ-phosphate, bringing it into an almost eclipsed position relative to the α-phosphate (*Figure 7*). In this conformation, the repulsion by the negatively charged oxygen atoms of α- and β-phosphates (*Cannon, 1993*) would push away the γ-phosphate group and promote hydrolysis. The rotation of γ-phosphate was accompanied by formation of a new hydrogen bond between the backbone nitrogen of Asn226 and $O^{2G}$. Elsewhere we show that the corresponding bond is seen in most structures of P-loop NTPases containing NDP-AlF$_4^-$ complex (*Shalaeva et al., 2018*), which is believed to be the closest analog of the transition state (*Jin et al., 2017b*). In contrast, this H-bond could be indentified only in few structures containing ADP-AlF$_3$ complexes or non-hydrolyzable NTP analogs. It appears that this bond contributes to the stabilization of the transition state during hydrolysis. In most P-loop NTPases the counterpart of Asn226 is a small residue, usually glycine or alanine, which might indicate the flexibility of the P-loop backbone in this position. In contrast, in MnmE GTPase, as well as in other $K^+$-dependent P-loop NTPases, this position is taken by an Asn or Asp residue that coordinates the activating cation, see *Figures 1B*, *6* and *7*. It appears that the activating cation not only directly interacts with the $O^{3G}$ atom of γ-phosphate (see *Figures 1B*, *6* and *7*), but also, indirectly, communicates with the $O^{2G}$ atom via the backbone nitrogen atom of Asn/Asp in the Asn226 position; this interaction may be functionally relevant.

Formation of new bonds between the $K^+$ ion and $O^{3G}$ as well as between Asn226 and $O^{2G}$ would lead to a reshuffling of the H-bond network around γ-phosphate, which is believed to be important for catalysis (*Wittinghofer and Vetter, 2011*; *Kamerlin et al., 2013*; *Jin et al., 2016*; *Jin et al., 2017a*; *Jin et al., 2017b*). The shape of the P-loop and the pattern of NTP binding to the P-loop are extremely well conserved across all P-loop NTPases (*Figure 6*), which indicates that activating entities could rotate γ-phosphate and promote novel stabilizing H-bonds also in P-loop NTPases of other families, as discussed elsewhere (*Shalaeva et al., 2018*).

The affinity of the AG site to the K$^+$ ion is intrinsically low (**Supplementary file 1B**, **Figure 2C**), therefore binding of K$^+$ ions to this site in M$^+$-dependent P-loop NTPases of the TRAFAC class requires a full-fledged K-loop, an extended version of the Switch I region, which provides additional ligands for the cation, see **Figures 1B**, **6** and **7** and (**Ash et al., 2012**). Still, while possessing the K-loop, most available structures of K$^+$-dependent NTPases do not contain a bound K$^+$ ion (**Figure 6—figure supplement 1**). The observed absence of K$^+$ ions from most structures of K$^+$-dependent P-loop NTPases (**Figure 6—figure supplement 1**) could be due to several reasons, including their absence from the crystallization medium as in the K$^+$-dependent GTPase Era, (**Figure 6—figure supplement 3**), where the potential K$^+$ binding site contains a water molecule surrounded by six potential K$^+$ ion ligands. Even when K$^+$ ions were present in the crystallization medium, the electron density difference between the K$^+$ ion (18 electrons) and the water molecule (10 electrons) is often insufficient to easily distinguish their relative contributions to the diffraction pattern (**Kuhle and Ficner, 2014**). Thus, at 60% occupancy, the K$^+$ ion cannot be distinguished from a water molecule (**Shui et al., 1998**). However, in most crystal structures of K$^+$-dependent GTPases (**Supplementary file 1A**), not only the M$^+$ ion is absent, but the entire K-loop is either unresolved or shows up far away from the active site (**Figure 6—figure supplement 1**). In the structures with an undefined position of the K-loop, the M$^+$-binding site is incomplete, although all the sequence features of an M$^+$-dependent protein, as defined by Ash et al (**Ash et al., 2012**), are present. Thus, additional factors appear to affect the K$^+$ ion binding.

One of such factors could be inferred from the comparison of crystal structures of the cation-dependent GTPases MnmE and Era in their active and inactive conformations. A full-fledged cation binding site was absent from the inactive conformations of MnmE (**Figure 6—figure supplement 2**) and Era (**Figure 6—figure supplement 3**), but present in the structures where they were crystallized together with their physiological activating partners. Notably, dimerization of the G-domains of MnmE required both the GTP nucleotide and K$^+$ ions in the medium, whereas Na$^+$ ions could not support dimerization, even in the presence of GTP (**Meyer et al., 2009**; **Böhme et al., 2010**). In the complex of Era with its activator, a 16S rRNA fragment (PDB: 3R9W), K$^+$ ions were missing because of their absence from the crystallization solution. Still, the K-loop attained the shape required for cation binding and the cation-binding site was complete, with all the coordination bond partners at short distances (<3.5 Å) from the water molecule that occupied the place of the K$^+$ ion (**Figure 6—figure supplement 3**).

The disordered K-loop in the inactive states of MnmE and Era and the stabilized K-loop in their active states suggest that the interaction with the activating partner stabilizes the functional, K$^+$-binding conformation of the K-loop, which enables binding of the K$^+$ ion and its subsequent interaction with the NTP molecule. Indeed, proper conformation of the K-loop (Switch I region) is crucial for the cation binding, since this loop provides two backbone oxygen atoms as ligands for the cation. We believe that the same mechanism could be involved in the activation of other K$^+$-dependent NTPases (**Table 2**), whereby the proper conformation of the K-loop and functionally relevant K$^+$ binding could be promoted by interaction with the activating protein or RNA/DNA partner.

In RecA-like recombinases (**Figure 6—figure supplement 4**), the K$^+$ ion in the AG site is coordinated by a conserved Asp residue, which is responsible for the K$^+$-dependent activation (**Qian et al., 2006**). This residue (Asp302 in PDB: 2F1H) is provided by the adjacent monomer within the RadA homooligomer that assembles upon interaction of RecA proteins with double-stranded DNA. Thus, in RecA-like recombinases, the K$^+$-binding sites differ from those in K$^+$ (or Na$^+$)-dependent TRAFAC NTPases, but, similarly to TRAFAC NTPases, appear to attain functionality upon the interaction with the activating partner that provides ligands for the K$^+$ ion.

## Evolutionary implications and the riddle of dynamins

The major classes of P-loop NTPases appear to have emerged before the divergence of bacteria and archaea (**Lupas et al., 2001**; **Leipe et al., 2002**; **Ponting and Russell, 2002**; **Söding and Lupas, 2003**; **Orengo and Thornton, 2005**; **Ranea et al., 2006**; **Alva et al., 2015**; **Wuichet and Søgaard-Andersen, 2015**; **Gogarten et al., 1989**; **Iwabe et al., 1989**; **Leipe et al., 2003**). An evolutionary scenario for the origin of P-loop NTPases has been recently proposed by Lupas and colleagues, who hypothesized that the ancestor of P-loop NTPases was an NTP-binding protein incapable of fast NTP hydrolysis, but, perhaps, involved in the transport of nucleotides (**Alva et al., 2015**). Indeed, as already discussed (**Leipe et al., 2002**), the main common feature of the P-loop NTPases is the

eponymous motif, which was identified as an antecedent domain segment by Lupas and colleagues (*Lupas et al., 2001*). Milner-White and coworkers argued that the very first catalytic motifs could have been short glycine-rich sequences capable of stabilizing anions (nests) (*Bianchi et al., 2012*; *Watson and Milner-White, 2002*) or cations (niches) (*Torrance et al., 2009*); such motifs can still be identified in many proteins. Specifically, the P-loop was identified as a nest for the phosphate group (s) (*Bianchi et al., 2012*; *Alva et al., 2015*). We showed here that the P-loop motif specifically imposes the same extended, catalytically-prone conformation on bound NTP molecules in different families of P-loop NTPases (*Figures 5–7*, *Figure 5—figure supplement 1*).

The conformational space of the Mg-NTP complex in water, as sampled by our MD simulations (*Figures 4* and *5*, *Figure 5—figure supplement 1*), reflects the preferred phosphate chain conformations in water and in the presence of monovalent cations. $K^+$ and $NH_4^+$ ions brought Mg-ATP into extended conformations that were most similar to the catalytically-prone conformations observed in the active sites of P-loop NTPases. It is tempting to speculate that the P-loop could have been shaped in $K^+$- and/or $NH_4^+$-rich, but $Na^+$-limited environments, which would favor the extended conformations of unbound (free) NTPs. Indeed, $Na^+$, the ion with the smallest diameter in this study, is known to exhibit the strongest binding to the phosphate chain, which has been reproduced in our MD simulations (*Supplementary file 1B*, *Figure 2*, *Figure 2—figure supplement 2*). Consequently, tightly bound $Na^+$ ions would keep the phosphate chain in a contracted/curled conformation in water (*Figures 4* and *5*, *Figure 5—figure supplement 1*). $K^+$ and $NH_4^+$ ions are larger and form longer coordination bonds, which results in the wider $P^B$-$O^{3B}$-$P^G$ angles and longer $P^A$-$P^G$ distances (*Table 1*, *Figures 4* and *5*, *Figure 5—figure supplement 1*). However, binding of $K^+$ and $NH_4^+$ ions to the phosphate chain is weaker than binding of $Na^+$ (*Figure 2C–E*, *Supplementary file 1B*). Thus, extended conformation of the phosphate chain in water could be reached in the presence of $K^+$ and/or $NH_4^+$ ions only if their concentrations were distinctly higher than those of $Na^+$ ions.

As argued by Lupas and colleagues, one of the possible mechanisms for the emergence of diverse classes of P-loop NTPases could be a combination of the same 'original' NTP-binding P-loop domain with different partners that could promote the insertion of an activating moiety into the active site (*Alva et al., 2015*). This suggests that $K^+$ ions and/or amino groups were available as activating cofactors during the emergence of P-loop NTPases. Hence, the P-loop motif itself may have been shaped by the high levels of $K^+$ and/or $NH_4^+$ ions in the habitats of the first cells. Since the emergence of the P-loop motif happened at the very beginning of life, when the ion-tight membranes were unlikely to be present, the match between the shape of the P-loop and large cations of $K^+$ and $NH_4^+$ is consistent with our earlier suggestions on the emergence of life in terrestrial environments rich in $K^+$ and nitrogenous compounds (*Mulkidjanian et al., 2012*; *Dibrova et al., 2015*).

The activating Arg/Lys residues are usually provided upon interactions of the P-loop with another domain of the same protein, or an adjacent monomer in a dimer or an oligomer, or a specific activating protein, or DNA/RNA (*Table 2*), so that this activation can be tightly controlled, see (*Shalaeva et al., 2018*) and references therein. For cation-dependent NTPases of the TRAFAC class, however, the situation is different: the cation-binding K-loop is an extended Switch I region of the same P-loop domain (*Figures 1B, C* and *6B*). If the formation of the K-loop and binding of an $M^+$ ion to it were able to proceed in an uncontrolled way, then the cell stock of ATP/GTP would be promptly hydrolyzed by constantly activated $M^+$-dependent NTPases. This, however, does not happen; $M^+$-dependent NTPases are almost inactive *in solo* and attain the ability to hydrolyze NTPs only after binding to an activating partner. This behavior is in line with our MD simulations that indicate rather poor binding of $K^+$ ions to the 'naked' AG site of the ATP molecule (*Figure 2C*, *Figure 2—figure supplements 1* and *2*). This poor $K^+$ binding manifests itself also in the need to use very high (>>100 mM) levels of potassium salts to activate the $K^+$-dependent P-loop NTPases in the absence of their physiological activating proteins or RNA (*Fasano et al., 1982*; *Kuhle and Ficner, 2014*). As our comparative structure analysis showed, the functional K-loop in such NTPases is distorted in the inactive (apo-) state (*Figure 6—figure supplement 1*), but attains its functional shape and eventually binds the cation upon the interaction with the activating partner (*Figure 6—figure supplements 2* and *3*). The interaction with the activator, however, must be highly specific to prevent the activation of hydrolysis in response to an occasional binding to a non-physiological partner. It indeed seems to be specific; *Table 3* lists structures of the eukaryotic translation initiation factor eIF5B in which a kind of a K-loop formed not via their functional interaction with the ribosome, but through non-physiological crystal-packing contacts (*Kuhle and Ficner, 2014*). Although these quasi-K-loops bind different

monovalent cations, the corresponding structures contain GTP molecules, indicating the absence of hydrolytic activity. In addition, the respective $P^A$-$P^G$ distances are shorter than those in the structures of P-loop NTPases in their active conformations (*Table 3*). Apparently, in addition to cation binding, some other factors may control the catalysis and prevent spurious NTP hydrolysis. Some of these factors are discussed elsewhere (*Shalaeva et al., 2018*).

The smaller Na$^+$-ion, while tightly binding to individual phosphate groups, can neither simultaneously reach the $O^{2A}$ and $O^{3G}$ atoms of the extended phosphate chain (*Figure 1C*), nor contract the extended phosphate chain because it is fixed by the P-loop, which explains why Na$^+$ ions are not competent in most P-loop NTPases. As mentioned in the Introduction, only eukaryotic dynamins can be activated both by K$^+$ and Na$^+$ ions (*Ash et al., 2012*; *Chappie et al., 2010*; *Yan et al., 2011*). How do small Na$^+$ ions activate GTP hydrolysis in dynamins? Dynamin-like proteins are activated upon dimerization, and crystal structures of their dimers in complex with GTP analogs and Na$^+$ ions are available (*Table 3*, *Figure 1C*, *Figure 8*). These structures contain fully resolved K-loops, which allowed us to compare the structures of the Na$^+$-adapted dynamin and K$^+$-dependent MnmE GTPase with the results of our MD simulations. In MD simulations, presence of Na$^+$ ions led to contracted phosphate chain conformations (*Figures 4* and *8A*), whereas crystal structures of dynamins showed extended conformations of the phosphate chain even with a Na$^+$ ion bound (*Figure 8B*). In the dynamin, the phosphate chain is in the catalytically prone extended conformation owing to its stabilization by the residues of the P-loop, so that the small Na$^+$ ion interacts with the γ-phosphate

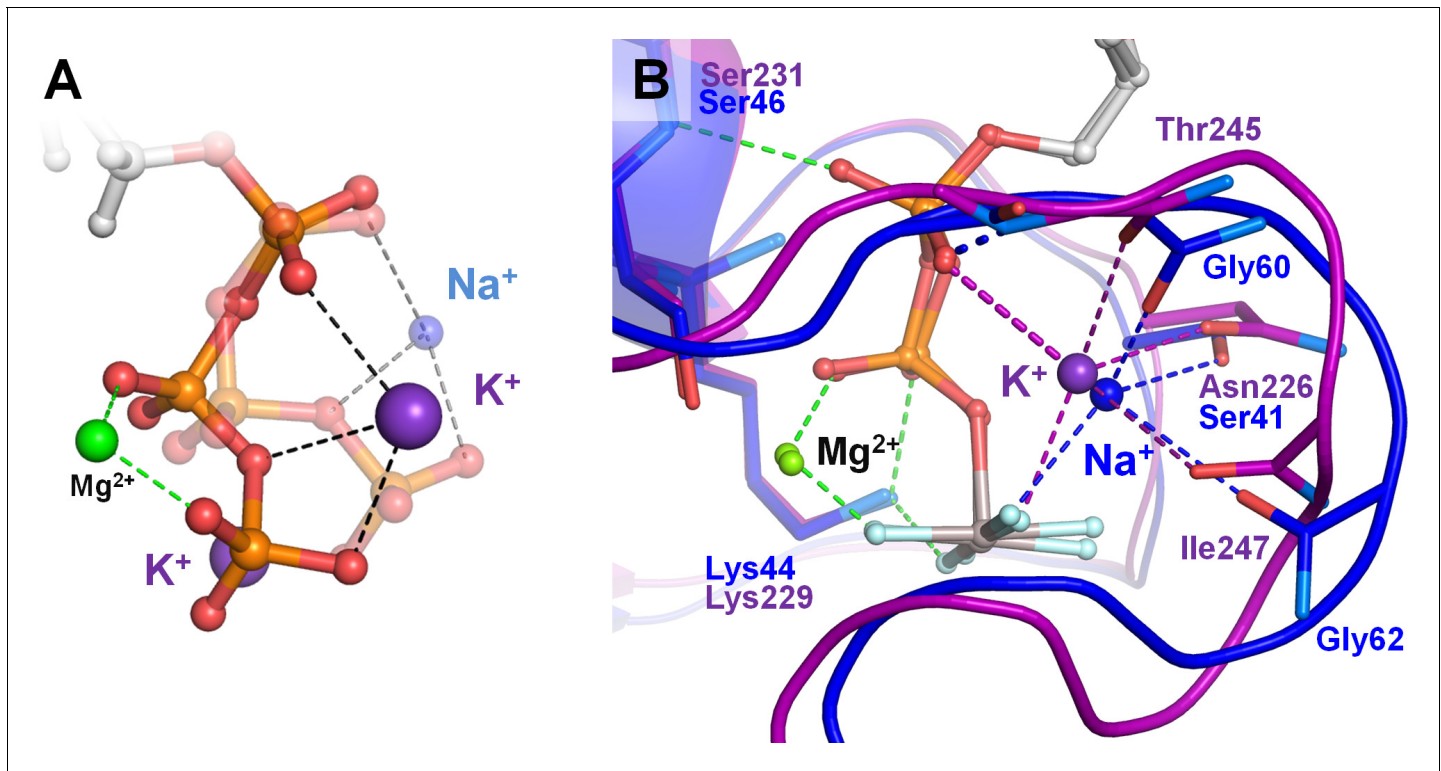

**Figure 8.** Effects of Na$^+$ binding on the shape of phosphate chain in solution and in Na$^+$-adapted P-loop NTPases. The color scheme is as in *Figure 1*, except that Al and F atoms in the GDP-AlF$_4^-$ complexes are colored grey and cyan, respectively. (**A**) Superposition of the K$^+$-bound (solid structure) and Na$^+$-bound (transparent structure) conformations of the triphosphate chain as obtained from MD simulations of an ATP molecule in water. Data from MD simulations 4–8 in *Supplementary file 1C*. (**B**) Superposition of the P-loop NTPase structures with a bound K$^+$ ion (MnmE GTPase, PDB: 2GJ8 (*Scrima and Wittinghofer, 2006*), purple) and Na$^+$ ion (dynamin, PDB: 2X2E (*Chappie et al., 2010*), blue). Proteins are shown as a cartoon. Dashed lines indicate hydrogen bonds and coordination bonds. Bonds that occur in all P-loop NTPases are shown in green, those that occur in K$^+$-binding proteins are in purple, those bonds that occur in Na$^+$-binding dynamin-like proteins are in blue. The thick dashed purple line indicates the bond between the K$^+$ ion and the oxygen atom of α-phosphate, which is absent in dynamins. The thick dashed blue line indicates the dynamin-specific H-bond between $O^{2A}$ atom and the backbone amide group of the shortened K-loop.
DOI: https://doi.org/10.7554/eLife.37373.027

but cannot reach the oxygen atom of the α-phosphate (*Table 3*, *Figure 8B*). The ability of dynamins to keep the $Na^+$ ion in the AG position appears to be due to several mutations (*Ash et al., 2012*), *cf Figure 1B and C* and see *Figure 8B*. Specifically, in dynamins, (i) the conserved Asn in the P-loop is replaced by a shorter Ser residue; (ii) the K-loop is shortened by one residue, and (iii) the Asn residue responsible for the K-loop conformation is replaced by the longer Glu residue. These mutations allow the K-loop to come closer to the $Na^+$ ion and stabilize it in the AG site by backbone carbonyl oxygens even in the absence of a bond between the $Na^+$ ion and $O^{2A}$ atom (*Figures 1C* and *8B*). In dynamin structures, the free $O^{2A}$ atom is coordinated by the backbone amide group of the shortened K-loop residue (Gly60 in the human dynamin PDB 2X2E (*Chappie et al., 2010*)). This interaction is not seen in the structures of $K^+$-dependent GTPases (*cf. Figure 1B* with *Figure 1C*). It seems that the additional coordination of $O^{2A}$ by the Gly residue of the shortened K-loop serves as a functional replacement of its coordination by the $K^+$ ion. The truncated K-loop appears to be flexible enough to accommodate either $K^+$ or $Na^+$ ions, allowing dynamins to be equally well activated by $K^+$ and $Na^+$ ions (*Chappie et al., 2010*; *Yan et al., 2011*). In the available $Na^+$-containing dynamin structures with the $GTP-AlF_4^-$ analog bound, the $P^A$- $O^{2A}$ and $P^G$-F bonds stay parallel (eclipsed) to each other and overlap with respective bonds in the structure of $K^+$-containing MnmE GTPase (*Figure 8B*). Apparently the described set of alternative interactions secures the ability of the $Na^+$ ion to promote hydrolysis by twisting the γ-phosphate group.

The adaptation to $Na^+$ ions required at least three mutational changes in the highly conserved parts of the protein, see (*Ash et al., 2012*) and *Figure 8*. The low probability of this combination of changes may explain why just this one case of $Na^+$-adaptation is known. In contrast, Arg/Lys residues are widespread as activators of P-loop NTPases, see *Table 2* and (*Shalaeva et al., 2018*). In a few cases (e.g. in TRAFAC class NTPases) it was possible to trace how Arg residues replaced $K^+$ ions in the course of evolution in different lineages (*Mulkidjanian et al., 2012*; *Dibrova et al., 2015*). The recruitment of an Arg/Lys residue as an activating moiety is relatively simple and makes the catalysis independent of the oscillations of $K^+$ and $Na^+$ levels in the cell.

Although $Na^+$ ions cannot activate most P-loop NTPases, their binding in the active sites may be physiologically relevant. Recently Gorfe and colleagues, while performing microsecond-scale MD simulations of oncogenic Ras GTPases, noted tight binding of $Na^+$ ions to the phosphates of GTP molecules bound to the P-loop (*Alva and Lupas, 2018*). Ras GTPases are activated not by $K^+$ ions, but by arginine fingers of their specific GTPase activating proteins (GAPs) (*Wittinghofer and Vetter, 2011*; *Bos et al., 2007*). Gorfe and colleagues suggested that such a $Na^+$ binding might prevent hydrogen bonding of Tyr32 of Ras with the γ-phosphate of GTP, which appears to prevail in the absence of the respective GAP (*Sayyed-Ahmad et al., 2017*). It is tempting to speculate that bound, but not activating $Na^+$ ion(s) may also hinder the access of the activating Arg or Lys fingers. In active cells, the cytoplasm contains much more $K^+$ ions than $Na^+$ ions (*Macallum, 1926*; *Williams and Frausto da Silva, 1991*). Binding of $K^+$ ions in the AG sites of $K^+$-independent NTPases must be weak and is unlikely to hinder the interaction with their activators. However, in energy- or nutrient-limited prokaryotic cells, for example in the stationary phase of growth, a reversal of the $[K^+]/[Na^+]$ ratio is observed (*Fagerbakke et al., 1999*) because cells cannot maintain the $[K^+]/[Na^+]$ disequilibrium on their membranes without an energy source. One would expect that energy-limited or energy-depleted cells would exhaust all their NTP stock and die. Instead, prokaryotic cells often go into a dormant state; this state is characterized by retention of an essential part of the NTP stock that is eventually used for awakening of dormant cells under more favorable conditions, see (*Mulkidjanian et al., 2012*; *Dibrova et al., 2015*) and references therein. Elsewhere we argued, that high $Na^+$ levels in dormant cells could suppress their metabolic activity by blocking the $K^+$-dependent NTPases (*Mulkidjanian et al., 2012*; *Dibrova et al., 2015*). However, most NTPases in the cell are activated not by $K^+$ions, but by Arg or Lys residues. The remarkable observations of Gorfe and colleagues suggest that these NTPases could also be hampered by $Na^+$ ions, which would then explain how dormant cells succeed to retain their NTP stock.

## Relation to NTPases with other folds

Our MD simulations of the behavior of an unconstrained Mg-ATP complex in water showed correlations between binding of cations to the ATP molecules and their conformation. The data obtained provide information not only on the interaction of $M^+$ ions with Mg-ATP complexes in the bidentate βγ coordination of the $Mg^{2+}$ ion, which is typical for the P-loop NTPases, but also on their interaction

with tridentate αβγ-coordinated Mg-ATP complexes (*Figures 4* and *5A*, *Figure 4—figure supplement 1*, *Table 1*).

The tridentate αβγ-coordination is found, for instance, in the K⁺-dependent chaperonin GroEL and related proteins. Unlike P-loop NTPases, the GroEL from *E. coli* and the related chaperonin Mm-cpn from *Methanococcus maripaludis* were inhibited by Na⁺ ions even when Na⁺ was added over K⁺ (*Kusmierczyk and Martin, 2003*). In the crystal structures of GroEL, K⁺ ion was identified in the position that corresponded to the AG site of our MD simulations, *cf* the right structure in *Figure 4B* with PDB: 1PQC (*Chaudhry et al., 2003*) or PDB: 1KP8 (*Wang and Boisvert, 2003*). The P$^A$-P$^G$ distance for the ATP analogs is 4.4 Å in the former and 4.3 Å in the latter structure. These distances are similar to the one obtained in the MD simulations for the tridentate αβγ-coordinated Mg-ATP complexes in the presence of K⁺ ions (4.32 ± 0.24 Å); in the presence of Na⁺ ions the distance was shorter, 4.26 ± 0.37 Å (*Table 1*). The available structures of Mm-cpn (PDB: 3RUV and 3RUW (*Pereira et al., 2012*)) contain only a water molecule in the AG position of the bound nucleotide; this water molecule, however, is surrounded by five oxygen atoms at <3 Å distance, indicating the presence of a typical cation-binding site.

For GroEL, K⁺ ions were shown to increase the affinity to the nucleotide (*Grason et al., 2008*). It appears that the phosphate chain, unlike those tightly bound to the P-loops, retains certain flexibility in GroEL-type ATPases, so that its shape depends on the size of the monovalent cation, as it was observed in our MD simulations. Here, binding of the Na⁺ ion would lead to a contracted, supposedly, less catalytically prone conformation. Thus, Na⁺ ions added over K⁺ ions, owing to their ability to bind more tightly, would inhibit ATP hydrolysis, in agreement with experimental observations (*Kusmierczyk and Martin, 2003*). The example of GroEL-type ATPases shows that the balance between compensating the negative charge of the triphosphate chain and maintaining its catalytically-prone conformation might be important not only for P-loop NTPases, but also for other NTPase superfamilies. Accordingly, our MD simulation data may help clarify the mechanisms in other NTPases.

## Conclusions

To address the mechanism of NTP hydrolysis in K⁺-dependent NTPases, we performed MD simulations of Mg-ATP and Mg-GTP complexes in water and in the presence of K⁺, Na⁺, or NH₄⁺ ions. These MD simulations revealed specific binding sites that were the same for all studied cations. Comparisons of the MD simulation data with crystal structures of P-loop NTPases showed that the tight cation-binding site between the β- and γ-phosphates, identified in the MD simulations, coincided with the position of the invariant lysine residue of the P-loop (Walker A) motif, whereas the loose binding site between the α- and γ-phosphates corresponded to the position occupied by the activating moiety - either the lysine/arginine finger or the activating M⁺ ion. In all analyzed structures, the P-loop motif keeps the triphosphate chains of enzyme-bound NTPs in a strictly conserved, extended, catalytically prone conformation, similar to that imposed on NTPs in water by large K⁺ or NH₄⁺ ions bound between the α- and γ-phosphates. MD simulations of K⁺-dependent GTPase MnmE showed that linking of the α- and γ-phosphates by the activating K⁺ ion led to the rotation of the γ-phosphate yielding an almost eclipsed, catalytically productive conformation of the triphosphate chain. The twisting of the γ-phosphate was accompanied by the formation of a new hydrogen bond between the backbone nitrogen atom of the K⁺-binding Asn226 residue of the P-loop motif and the O$^{2G}$ atom. Because of their smaller size and shorter coordination bonds, Na⁺ ions cannot bind between the α- and γ-phosphates when the phosphate chain is stretched by the P-loop, which could explain their inability to activate most P-loop NTPases.

In the largest TRAFAC class of P-loop NTPases, K⁺-dependent NTPases appear to precede in evolution those enzymes that are activated by Arg or Lys fingers (*Mulkidjanian et al., 2012*; *Dibrova et al., 2015*). Therefore, twisting of the γ-phosphate by the activating ion and formation of an additional stabilizing hydrogen bond between the P-loop and the γ-phosphate might represent the basic mechanism of P-loop NTPases. Currently, most P-loop NTPases are activated by Arg/Lys fingers of their physiological counterparts, which makes the catalysis independent of the oscillations of K⁺ and Na⁺ levels in the cytoplasm. The few K⁺-dependent NTPases are mostly involved in translation and/or in interaction with RNA. It is tempting to speculate that these enzymes, involved in the most ancient and most slowly evolving parts of the cellular machinery, could retain the ancestral K⁺-

dependence because RNA structures are surrounded by 'clouds' of $K^+$ ions that compensate the negative charges of RNA backbone phosphate groups (*Draper, 2004*).

In cation-dependent P-loop NTPases, the very formation of the $M^+$-binding site next to the P-loop appears to require additional interactions with activating domains or proteins. This trait prevents uncontrolled depletion of the cellular NTP stock.

## Materials and methods

### Molecular Dynamics simulations

To investigate the effects of cation binding on the structure of the Mg-ATP and Mg-GTP complexes, we have conducted free MD simulations of Mg-NTP complexes in water solution alone and in the presence of $K^+$, $Na^+$, or $NH_4^+$ ions. Together with monovalent cations, $Cl^-$ ions were added to balance the total charge of the system. For the simulation of Mg-NTP complexes in water solution without additional ions, two positive charges had to be added to balance the total charge of the system. We added two dummy atoms with single positive charges and applied positional restraints to fix the positions of these atoms in the corners of the unit cell. In all systems, the ATP or GTP positions were restrained to the center of the cell by applying harmonical positional restraints to the $N_1$ atom of the nucleobase.

To investigate the effects of cation binding in the GTPase MnmE we conducted MD simulations of this protein in complex with Mg-GTP. We had modeled the following three states: (1) the active state with $K^+$ that was represented by the dimer of G-domains of MnmE, each with a $K^+$ ion bound (PDB: 2GJ8, resolution 1.7 Å, source: *E.coli*), (2) anthe (inactive) MnmE monomer with the $K^+$ ion replaced by a water molecule and a full-fledged K-loop (PDB:2GJ8, resolution 1.7 Å, source: *E.coli*), and (3) the inactive state represented by a G-domain with a distorted K-loop (PDB: 3GEI, resolution 3.4 Å, source: *Chlorobium tepidum*) The missing K-loop in the 3GEI PDB entry was modeled using Modeller 9v7 for the simulation of inactive MnmE. In each case $K^+$ and $Cl^+$ ions were added to surrounding media.

For simulations, we used CGenFF v.2b8 parameters for $ATP^{4-}$, $GTP^{4-}$ and $NH_4^+$ molecules, an extension of the CHARMM force field designed for small molecules (*Vanommeslaeghe et al., 2010*). For simulations of Mg-NTP in water we used the TIPS3P water model, which differs from other classical models in the presence of additional van der Waals parameters for interactions between water molecules (*Jorgensen et al., 1983*). For simulations of P-loop NTPases, we used the classical CHARM36 force field with water model TIP3P (*Vanommeslaeghe et al., 2010*). For the $Mg^{2+}$ ion, we used parameters designed by Callahan et al. (*Callahan et al., 2010*). For $Na^+$ and $K^+$ ions, we used parameters of Joung and Cheatham (*Joung and Cheatham, 2008*).

Non-bonded interactions were computed using particle mesh Ewald method with 10 Å real space cutoff for electrostatic interactions and the switching functions between 10 and 12 Å for the van der Waals interactions. The multiple time-step method was employed for the electrostatic forces; the non-bonded interaction list was constructed using a cutoff of 14 Å and updated every 20 steps. The covalent bonds involving hydrogen atoms were constrained using the SHAKE algorithm (*Ryckaert et al., 1977*) (the MD integration step, one fs). Then the water box and ions were added; after the addition of $Na^+$ or $K^+$ and neutralizing ions the total ionic strength was 0.2 M.

Molecular dynamics simulations were performed in the *NPT* ensemble. Temperature was maintained at $T=$ 298 K with the Berendsen thermostat using a coupling parameter of 5 $ps^{-1}$(*Berendsen et al., 1984*). The pressure was maintained at one atm by the Langevin piston method with the piston mass of 100 amu and Langevin collision frequency of 500 $ps^{-1}$ (*Feller et al., 1995*).

Each system was first optimized through energy minimization followed by a 20 ns equilibration run. For Mg-ATP complex, free MD simulations were conducted for 170 ns in three independent runs (500 ns total) for each of the four systems ($K^+$, $Na^+$, $NH_4^+$ and no extra ions). Additional series of 20–25 ns simulations were conducted for both Mg-ATP and Mg-GTP complexes. For the MnmE GTPases, each system was simulated for 100 ns (*Supplementary file 1C*).

In our calculations, we used Gromacs v.4.5.5 (*Pronk et al., 2013*) software with MPI implementation at the supercomputer SKIF 'Chebyshev' at the Computational Center, Moscow State University.

Analyses of MD data were performed with MATLAB software (The Mathworks, Inc.). The VMD software (*Humphrey et al., 1996*) was used for visualization of the MD simulations results.

## Statistical analysis

To analyze conformations of Mg-ATP complexes in the presence of different cations, we selected MD simulation fragments of similar lengths with the same type of interactions between the $Mg^{2+}$ ion and the triphosphate chain. In each case ~160 ns of MD simulation were taken to characterize a particular Mg-ATP conformation; if needed, results of several independent simulations were merged to collect enough data, see *Figure 4—figure supplement 2A,B* and *Figure 4—figure supplement 3A, B* for examples. For the MD simulation data, we calculated autocorrelation functions (*Figure 4—figure supplement 2C* and *Figure 4—figure supplement 3C*). Given the correlation times obtained, independent frames were extracted to calculate characteristic values for the separate conformations of ATP. For the systems without additional monovalent cations, every N-th frame was taken for the calculation, with N defined by the correlation time. For the systems with monovalent cations, only frames in which at least one monovalent cation was bound to the phosphate chain were taken, with at least N frames between measurements. A monovalent cation was considered to be bound when it was within a binding distance from at least one oxygen atom of the phosphate chain, with binding distances defined as follows: 2.4 Å for $Na^+$ and 3.2 Å for $K^+$ and $NH_4^+$.

To compare the conformations of Mg-ATP, as obtained in MD simulations with different monovalent cations, we used the two-sample t-test. We used the assumption that the two compared data samples were from populations with equal variances; the test statistics under the null hypothesis had Student's t distribution with *n + m–2* degrees of freedom, where *n* and *m* were sample sizes, and the sample standard deviations were replaced by the pooled standard deviation. In each case, the null hypothesis was that the data in the two samples come from independent random samples with normal distributions with equal mean values and equal but unknown variances. The alternative hypothesis was that the data in the two samples come from populations with unequal mean values. The test rejects the null hypothesis at the 5% significance level. In *Supplementary file 1G-1L* we compare the test results and particular P-values obtained for all pairwise comparisons conducted in this study. Statistical analysis of MD trajectories was performed with MATLAB software (The Mathworks, Inc.).

## Protein structure analysis

For statistical analysis of the PDB structures, we used the InterPro database (*Finn et al., 2017*). A list of PDB IDs of P-loop proteins was extracted for the InterPro entry IPR027417 and filtered with the RCSB PDB search engine (*Rose et al., 2017*) to include only those structures that contained $Mg^{2+}$ ion and one of the following molecules (in RCSB PDB chemical IDs): ATP, GTP, ANP, GNP, ACP, GCP, ASP, GSP, ADP, and GDP. We used MATLAB software (The Mathworks, Inc.) to measure the distances from the NTPs (or their analogs) to the surrounding Lys/Arg residues and selected only those structures with the nucleotide bound to at least one Lys (indicating that the nucleotide is indeed bound to the P-loop and the P-loop Lys residue is not mutated). MATLAB software (The Mathworks, Inc.) was also used to measure the shape of the phosphate chain in each NTP-like substrate or the transition state-mimicking molecule.

To characterize $M^+$-binding sites of P-loop proteins, we have searched the available literature data for cation-dependent activities of the respective proteins, with the results summarized in *Supplementary file 1A*. For each of those proteins, we have examined the available crystal structures in order to characterize the cation binding site(s). In total, we have selected 17 structures with metal cations, ammonium ions or water molecules (*Table 3*). Multiple superpositions of the P-loop proteins were built in PyMOL (*DeLano, 2010*) by matching coordinates of the P-loop motif together with the β-strand and α-helix flanking this loop using the PyMOL's 'super' function. Each protein was superposed onto the reference structure of the MnmE GTPase structure (PDB: 2GJ8) (*Scrima and Wittinghofer, 2006*). In addition to cation-dependent P-loop proteins we have chosen six cation-independent proteins from different families for comparison (*Figure 7C*).

## Acknowledgments

Very useful discussions with Drs. K Gerwert, A Gorfe, J Klare, EV Koonin, VP Skulachev and H-J Steinhoff are greatly appreciated. We are thankful to Dr. D V Dibrova and A A Mulkidzhanyan for their help during the launching phase of this project. The calculations were carried out using the equipment of the shared research facilities of HPC computing resources at Lomonosov Moscow State University supported by its Development Program and the project RFMEFI62117 × 0011 (AVG, DAC). This study was supported by the Deutsche Forschungsgemeinschaft, Federal Ministry of Education and Research of Germany, the EvoCell Program and Open Access Publishing Fund of the Osnabrueck University (AYM), the German Academic Exchange Service and the 14-50-00029 grant from the Russian Science Foundation (DNS). MYG is supported by the Intramural Research Program of the NIH at the National Library of Medicine.

## Additional information

### Funding

| Funder | Grant reference number | Author |
|---|---|---|
| Deutscher Akademischer Austauschdienst | Ostpartnerschaften programm | Daria N Shalaeva |
| Russian Science Foundation | 14-50-00029 (Noah's Ark) | Daria N Shalaeva Dmitry A Cherepanov Andrey V Golovin |
| U.S. National Library of Medicine | Intramural Research Program | Michael Y Galperin |
| Lomonosov Moscow State University | RFMEFI62117X0011 | Andrey V Golovin |
| Deutsche Forschungsgemeinschaft | DFG-436-RUS 113/963/0-1 | Armen Y Mulkidjanian |
| Bundesministerium für Bildung und Forschung | Laseromix | Armen Y Mulkidjanian |
| Osnabrueck University | EvoCell Program | Armen Y Mulkidjanian |

The funders had no role in study design, data collection and interpretation, or the decision to submit the work for publication.

### Author contributions

Daria N Shalaeva, Data curation, Software, Formal analysis, Investigation, Visualization, Methodology, Writing—original draft, Writing—review and editing; Dmitry A Cherepanov, Resources, Data curation, Software, Formal analysis, Validation, Investigation, Methodology, Writing—original draft, Writing—review and editing; Michael Y Galperin, Resources, Data curation, Software, Validation, Investigation, Methodology, Writing—original draft, Writing—review and editing; Andrey V Golovin, Software, Formal analysis, Investigation, Methodology, Writing—review and editing; Armen Y Mulkidjanian, Conceptualization, Supervision, Funding acquisition, Validation, Investigation, Methodology, Writing—original draft, Project administration, Writing—review and editing

### Author ORCIDs

Daria N Shalaeva (ID) http://orcid.org/0000-0003-0582-2612
Dmitry A Cherepanov (ID) http://orcid.org/0000-0001-6286-4638
Michael Y Galperin (ID) http://orcid.org/0000-0002-2265-5572
Andrey V Golovin (ID) http://orcid.org/0000-0002-8908-4268
Armen Y Mulkidjanian (ID) http://orcid.org/0000-0001-5844-3064

### Decision letter and Author response

Decision letter https://doi.org/10.7554/eLife.37373.033
Author response https://doi.org/10.7554/eLife.37373.034

## Additional files

### Supplementary files

• Supplementary file 1. (**A**) Monovalent cation requirements of P-loop GTPases and ATPases. (**B**) Properties of monovalent cations and their interactions with the $Mg^{2+}$-ATP complex. (**C**) Molecular dynamics simulations performed in this work. (**D**) Values of dihedral angles of the phosphate chains of Mg-ATP in the presence of $K^+$ ions. (**E**) Lifetimes of the βγ-conformation of Mg-ATP complex during MD simulations. (**F**) Characteristics of the triphosphate chain for different interactions between the $Mg^{2+}$ ion and ATP. (**G**) Comparison of the $P^A$-$P^G$ distance measurements of the βγ-coordinated Mg-ATP complexes. (**H**) Comparison of the $P^A$-$P^G$ distance measurements of the αβγ-coordinated Mg-ATP complexes. (**I**) Comparison of the $P^A$-$P^G$ distance measurements for the αβγ-coordinated and 'curled' βγ-coordinated Mg-ATP complexes in different systems. (**J**) Comparison of the $P^B$-$O^{3B}$-$P^G$ angle measurements for the βγ-coordinated Mg-ATP complexes. (**K**) Comparison of the $P^B$-$O^{3B}$-$P^G$ angle measurements for the αβγ-coordinated Mg-ATP complexes. (**L**) Comparison of the $P^A$-$P^G$ distance measurements for the αβγ-coordinated and 'curled' βγ-coordinated Mg-ATP complexes.
DOI: https://doi.org/10.7554/eLife.37373.028

• Transparent reporting form
DOI: https://doi.org/10.7554/eLife.37373.029

### Data availability

As obtained from MD simulations, we provide the structures of Mg-ATP complexes with bound $K^+$, $Na^+$ or $NH_4^+$ ions, as well as the structures of the G-domains of MnmE GTPases with and without activating potassium ions as source data files. Simulation data sets have been uploaded to Zenodo (https://zenodo.org/record/1888492#.XAasVhP7RTY).

The following dataset was generated:

| Author(s) | Year | Dataset title | Dataset URL | Database and Identifier |
|---|---|---|---|---|
| Shalaeva DN, Cherepanov DA, Galperin MY, Golovin AV, Mulkidzhanyan AY | 2018 | Molecular Dynamics simulation data for the article | https://zenodo.org/record/1888492#.XAasVhP7RTY | Zenodo, 10.5281/zenodo.1888492 |

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
