## [Decision Letter]

Thank you for submitting your article "Evolution of cation binding in the active sites of P-loop nucleoside triphosphatases" for consideration by *eLife*. Your article has been reviewed by two peer reviewers, and the evaluation has been overseen by a Reviewing Editor and Michael Marletta as the Senior Editor. The reviewers have opted to remain anonymous.

The reviewers have discussed the reviews with one another and the Reviewing Editor has drafted this decision to help you prepare a revised submission.

Summary:

This manuscript addresses an important and interesting question: in spite of a wealth of information available on NTPases it is still largely unclear how these enzymes are activated. These proteins constitute a large fraction of cellular enzymes and play a number of key roles in mechano-chemical events. Hence understanding molecular mechanisms of their activation, in particular by monovalent cations, is fundamental for our understanding of many important biological processes.

Shalaeva et al. describe the results of detailed molecular dynamics (MD) simulation analyses of solvated Mg-ATP in the presence and absence of potassium, sodium or ammonium ion, and compare the data with that observed in Mg-ATP/GTP bound to various P-loop NTPases. The major conclusion is that potassium and ammonium ions interact with the α and γ oxygens of the nucleoside to stabilize Mg-ATP in an extended conformation similar to that observed in P-Loop NTPases. Sodium was found to be too small to simultaneously interact with the α and γ oxygen atoms of a protein-bound extended phosphate chain. These observations help explain why some P-loop NTPases can be activated only by potassium and not sodium.

Overall, the work is detailed and the analysis rigorous; it is presented clearly and logically. Reviewers appreciated the attention to detail as well as the extensive citations accompanying a very useful Introduction and thoughtful Discussion. The comparative analysis of the role of Lys/Arg residues (such as from GAP proteins or another domain) with those of potassium and sodium is instructive.

We found that the manuscript lacks a couple of important pieces of information, and may benefit from some clarifications.

Essential revisions:

1) The major question deals with the objective difficulties in distinguishing monovalent cations from water molecules in crystal structures, due to a rather similar number of electrons and very weak anomalous signals produced by Na^+^ and K^+^. And although Na^+^ could in principle be distinguished owing to shorter coordinating distances, coordinating distances for the potassium ion are a perfect match to hydrogen bonding with molecules of water. Owing to this, reliable assignments can be made only on the basis of high resolution crystallographic structures (<1.8Å or so) and even then it would be advisable further validating such assignments through careful inspection of crystallographic data. In this regard, was resolution of crystals structures taken into account when selecting PDB entries for analysis and have crystal data been validated? Also, considering the availability of a large number of structures for NTPases, we wonder if there are cases where high resolution structures have been determined for both K^+^ and Na^+^ complexes? And if so, how these data (after appropriate validation) compare with the results of MD simulations presented in this paper?

2) The other issue involves MD simulation of Mg-ATP bound to NTPases. Despite the useful comparison of the MD data in solution conducted on the isolated Mg-ATP with the protein-bound nucleoside in X-ray structures, the two are clearly in very different conditions and therefore conclusions drawn from the comparative analysis are somewhat indirect. In addition, as the authors discuss in multiple places including the subsection “Activation of P-loop NTPases by monovalent cations”, crystallization conditions prevented observation of the ion or the K-loop entirely, thus making comparison difficult. This can be remedied by conducting MD on nucleoside-NPTase complexes; for at least one or two systems. This can be done reasonably easily. For example, at least in RAS GTPases, metal ions from bulk solvent readily bind to the phosphate group during MD simulations (e.g., https://www.ncbi.nlm.nih.gov/pubmed/28498561).

3) The third concern is that all of the simulations were conducted on ATP while most of the protein structure analysis involved GTPases. It appears that the underlying assumption is that ATPases and GTPases are mechanistically the same or at least similar. This may well be the case, and appears to be the subject of a separate article, but it would be helpful to the reader if the authors provided evidence to this effect. For example, conducting a few simulations of Mg-GTP in the presence of sodium and potassium should be straightforward.

4) Throughout the manuscript both "extended" and "eclipsed" conformations of the nucleoside are described as hydrolysis prone, although the latter is probably more prone. It would be helpful to somehow make a distinction.

5) Based on their work and some previous reports, such as the article cited above, it would seem that the nucleotide binding site of P-loop NTPases is susceptible to binding to monovalent metal ions, at least in the absence of compensatory interactions such as with GAPs. This raises the question as to whether a certain fraction of all NTPases may exist in a monovalent metal bound form in the cell. Authors' comment on this issue would be valuable.

6) Final, and somewhat minor, concern is a "diluted" end of the paper (Discussion section). We would encourage authors to add a clear set of brief conclusions at the very end, summarizing their findings and views on the role of monovalent cations and Arg/Lys fingers in the catalysis and the rationale for choosing one particular option for activation.

7) Avoid referring to the second "submitted" paper (Shalaeva et al.), because this paper remains unpublished and you wished to consider its submission separately.

---

## [Author Response]

Essential revisions:1) The major question deals with the objective difficulties in distinguishing monovalent cations from water molecules in crystal structures, due to a rather similar number of electrons and very weak anomalous signals produced by Na^+^ and K^+^. And although Na^+^ could in principle be distinguished owing to shorter coordinating distances, coordinating distances for the potassium ion are a perfect match to hydrogen bonding with molecules of water. Owing to this, reliable assignments can be made only on the basis of high resolution crystallographic structures (<1.8Å or so) and even then it would be advisable further validating such assignments through careful inspection of crystallographic data. In this regard, was resolution of crystals structures taken into account when selecting PDB entries for analysis and have crystal data been validated? Also, considering the availability of a large number of structures for NTPases, we wonder if there are cases where high resolution structures have been determined for both K^+^ and Na^+^ complexes? And if so, how these data (after appropriate validation) compare with the results of MD simulations presented in this paper?

We fully agree with the reviewer that, generally, distinguishing between monovalent cations and water molecules in structural data could be difficult. In the particular case of K^+^-dependent P-loop NTPases, however, this does not cause much of a problem because the number of respective structures is very small (Table 3). Although we have considered all the available structures of P-loop proteins with experimentally established dependence on monovalent cations, only few of them contained a cation in the active site (Table 3). In fact, we have not selected the PDB structures for the analysis; Table 3 contains all the available structures that contain both an NTP analog with a group that mimics gammaphosphate and a monovalent cation. In these structures, the nature of the cation bound is typically determined by the chemical nature of the monovalent cation in the crystallization medium, albeit we cannot exclude that in some cases a water molecule could be mistaken for a cation. The situation with highly resolved, reliable structures, as mentioned by the reviewer, is even worse. To date, there are only five structures with resolution better than 1.8 Å that contain a K^+^ ion and an NTP analog with a group that mimics γ-phosphate. Three of those structures do not belong to P-loop NTPases; the recently deposited structure of Pif1 helicase (PDB 5FTB) contains a K^+^ ion, but not in the active site that is taken by an Arg finger. The only remaining suitable structure is that of MnmE GTPase that we discuss and model in this work. In most cases, as we found, the cation in the active site could not be observed because the cation-binding loop was turned away from the nucleotide-binding site. We discuss the reasons for this absence of a bound cation in detail in the Discussion section.

2) The other issue involves MD simulation of Mg-ATP bound to NTPases. Despite the useful comparison of the MD data in solution conducted on the isolated Mg-ATP with the protein-bound nucleoside in X-ray structures, the two are clearly in very different conditions and therefore conclusions drawn from the comparative analysis are somewhat indirect. In addition, as the authors discuss in multiple places including the subsection “Activation of P-loop NTPases by monovalent cations”, crystallization conditions prevented observation of the ion or the K-loop entirely, thus making comparison difficult. This can be remedied by conducting MD on nucleoside-NPTase complexes; for at least one or two systems. This can be done reasonably easily. For example, at least in RAS GTPases, metal ions from bulk solvent readily bind to the phosphate group during MD simulations (e.g., https://www.ncbi.nlm.nih.gov/pubmed/28498561).

We are grateful to the reviewer for drawing our attention to the very interesting recent paper of Sayyed-Ahmad et al. We found this work to be very important for understanding the evolution of P-loop NTPases and discuss it at length in the revised manuscript. Although we performed new MD simulations of the Ras GTPase, because of the amount of the material involved, we have decided to present them in a separate paper that focuses on P-loop NTPases that are activated by Arg/Lys fingers, such as Ras-like proteins.

We also agree with the reviewer that MD data on isolated Mg-ATP complex cannot be directly applied to enzyme-NTP complexes. Following the suggestion of the reviewer, we performed, in collaboration with Dr. Andrey Golovin (included in the author list of the revised version), several additional MD simulations to analyze the interaction of K^+^ ions with nucleotide-NTP complexes in the K^+^-dependent GTPase MnmE, the only K^+^-dependentNTPase for which both K^+^- bound (activated) and K^+^-free (inactive) X-ray structures are available. Specifically, we have carried out MD simulations of the P-loop domains with a bound K^+^ ion (activated state) and without K^+^ (inactive state, two different systems were modeled) The results of these MD simulations indicate correlation between binding of the activating K^+^ ion in the space between α- and γ-phosphates of an extended triphosphate chain and twisting of the γ-phosphate group.

3) The third concern is that all of the simulations were conducted on ATP while most of the protein structure analysis involved GTPases. It appears that the underlying assumption is that ATPases and GTPases are mechanistically the same or at least similar. This may well be the case, and appears to be the subject of a separate article, but it would be helpful to the reader if the authors provided evidence to this effect. For example, conducting a few simulations of Mg-GTP in the presence of sodium and potassium should be straightforward.

We fully agree with the reviewer that simulations with a Mg-GTP complex would be useful in the given context provided that the majority of K^+^-dependent NTPases are GTPases. Therefore we have repeated all MD simulations of ATP in water also with GTP (in the absence of monovalent cations and in the presence of K^+^, Na^+^, and NH_4_^+^, respectively). The results were almost indistinguishable from the Mg-ATP data. We briefly mention these simulations throughout the main text of the revised manuscript and provide the respective plots for Mg-GTP in supplementary information (Figure 2—figure supplement 3 and Figure 5—figure supplement 1).

4) Throughout the manuscript both "extended" and "eclipsed" conformations of the nucleoside are described as hydrolysis prone, although the latter is probably more prone. It would be helpful to somehow make a distinction.

We thank the reviewer for this comment. In the revised manuscript, we consequently use the term "catalytically prone" for the conformation of a P-loop-bound extended NTP molecule in the absence of activator and "catalytically productive" for the extended AND near-eclipsed conformation of NTP in the presence of an activating moiety.

5) Based on their work and some previous reports, such as the article cited above, it would seem that the nucleotide binding site of P-loop NTPases is susceptible to binding to monovalent metal ions, at least in the absence of compensatory interactions such as with GAPs. This raises the question as to whether a certain fraction of all NTPases may exist in a monovalent metal bound form in the cell. Authors' comment on this issue would be valuable.

We thank the reviewer for this insightful comment. We have added the discussion of the possible interference of monovalent cations with the function of K^+^independent, Arg/Lys-activated GTPases in the Discussion section of the revised manuscript.

6) Final, and somewhat minor, concern is a "diluted" end of the paper (Discussion section). We would encourage authors to add a clear set of brief conclusions at the very end, summarizing their findings and views on the role of monovalent cations and Arg/Lys fingers in the catalysis and the rationale for choosing one particular option for activation.

Done.

7) Avoid referring to the second "submitted" paper (Shalaeva et al.), because this paper remains unpublished and you wished to consider its submission separately.

We have deposited the second paper by Shalaeva et al. to bioRxiv (preprint 439992). In the revised manuscript we provide references to the deposited manuscript.